# Causal Emotion Recognition in Conversation: Context Saturation and Discourse-Marker Evidence

**Cheonkam Jeong**
*University of California, Irvine*
*cheonkamjeong@gmail.com*
**Adeline Nyamathi**
*University of California, Irvine*

**Reviewed on OpenReview:** *https://openreview.net/forum?id=zCFQiJT7XN*

## Abstract

Despite strong recent progress in Emotion Recognition in Conversation (ERC), two gaps remain: we still lack a clear understanding of which modeling choices materially affect performance, and we have limited linguistic analysis that connects recognition findings to interpretable discourse-level patterns. We address both gaps via a systematic study on IEMOCAP, with a cross-dataset validation on MELD that supports the saturation framing while clarifying which effects are corpus-specific.

For recognition, we conduct controlled ablations with 10 random seeds and paired tests over seeds (with correction for multiple comparisons), yielding three findings. First, conversational context is the dominant factor: performance saturates quickly, with roughly 90% of the gain observed within our context sweep achieved using only the most recent 10–30 preceding turns (depending on the label set). Second, hierarchical sentence representations are most useful in utterance-only settings, with a clear advantage on MELD, but the benefit vanishes once turn-level context is available, suggesting that conversational history subsumes much of the intra-utterance structure. Third, a simple integration of an external affective lexicon (SenticNet) does not improve results, consistent with pretrained encoders already capturing much of the affective signal needed for ERC. Under a strictly causal (past-only) setting, our simple models attain strong performance (82.69% 4-way; 67.07% 6-way weighted F1), indicating that competitive accuracy is achievable without access to future turns.

For linguistic analysis, we examine 5,286 discourse-marker occurrences and find a reliable association between emotion and marker position within the utterance ($p < .0001$). In particular, *Sad* utterances show reduced left-periphery marker usage (21.9%) relative to other emotions (28–32%), aligning with accounts that link left-periphery markers to active discourse management. This pattern is consistent with our recognition results, where *Sad* benefits most from conversational context (+22%p), suggesting that sadness may be more context-dependent in this corpus than emotions with stronger local pragmatic cues.

**Code availability.** The code used for the experiments and analyses in this paper is available at `https://github.com/philhelenina/causal-erc-context-saturation`.

## 1 Introduction

Emotion Recognition in Conversation (ERC) is a key capability for socially intelligent dialogue systems, mental-health support tools, and empathetic conversational agents. Unlike sentence-level emotion classification, ERC requires models to interpret an utterance in relation to conversational history, speaker dynamics, and pragmatic signals that are often subtle or indirect. Recent text-only approaches built on large pretrained encoders have achieved strong performance on benchmarks such as IEMOCAP (Dutta & Ganapathy, 2024),

but progress has been accompanied by a growing mismatch between empirical gains and interpretability: it remains unclear which architectural ingredients actually matter and what linguistic evidence these models are exploiting.

**Gap 1: Which modeling choices materially matter in modern ERC?**  The ERC literature has introduced increasingly elaborate components—hierarchical encoders (Majumder et al., 2019), graph/knowledge-based modules (Zhong et al., 2020; Ghosal et al., 2019), lexicon fusion (Tu et al., 2022a), and sophisticated context modeling (Ghosal et al., 2019). However, many reported improvements rely on single-seed runs and heterogeneous experimental setups, making it difficult to separate robust effects from variance or configuration-specific artifacts. As a result, foundational design questions remain unresolved. How much conversational context is truly necessary before gains saturate? Does modeling intra-utterance sentence structure still help once turn-level context is available? Do external affective lexicons add signal beyond what pretrained encoders already capture?

**Gap 2: Recognition rarely connects to interpretable discourse-level patterns.**  High recognition accuracy does not automatically reveal how emotions are expressed in discourse, nor does it surface the linguistic structures that models implicitly exploit. In linguistics, discourse markers are well known to encode stance, subjectivity, and intersubjectivity (Schiffrin, 1987; Beeching & Detges, 2014). Their distribution at the left and right peripheries of utterances is linked to discourse management and listener-oriented effects (Fraser, 1999; Traugott, 2010; Beeching & Detges, 2014). Yet ERC research has rarely examined whether such positional patterns vary systematically by emotion. Connecting recognition results to discourse-marker positioning therefore complements predictive analysis with interpretable hypotheses about what aspects of conversational discourse contribute to emotional expression. We treat this analysis as descriptive linguistic evidence; whether such patterns can support emotion-conditioned generation is a separate empirical question that we leave to future work.

## 1.1 Research Questions

We address these gaps through a systematic study on IEMOCAP that combines controlled ablation experiments with large-scale discourse-marker analysis. Our investigation is guided by four research questions:

*Recognition: What architectural choices matter?*

**RQ1** Does conversational context improve emotion recognition, and how much context is sufficient before performance saturates?

**RQ2** Does hierarchical sentence representation help beyond flat utterance encoding, and under what context conditions?

**RQ3** Does incorporating an external affective lexicon (SenticNet) help under a simple integration scheme?

*Linguistic Analysis: What patterns exist?*

**RQ4** Are there emotion-specific discourse-marker positioning patterns in conversational discourse?

## 1.2 Contributions

1. We provide a statistically grounded ablation framework for ERC by reporting 10-seed evaluations with paired significance testing and correction for multiple comparisons, enabling reliable component-level conclusions.

2. We isolate the relative contributions of context length, hierarchical structure, and lexicon fusion (Tu et al., 2022a), showing that conversational context dominates and saturates quickly, while hierarchical encoding primarily benefits utterance-only settings.

3. We show that emotions differ substantially in their dependence on conversational history, with *sad* benefiting more from context than *angry*, suggesting that context effects are not fully explained by arousal alone.

4. We connect recognition to linguistic evidence by analyzing discourse-marker usage and positioning (Schiffrin, 1987; Fraser, 1999; Beeching & Detges, 2014; Traugott, 2010), showing that *Sad* utterances exhibit a small but robust reduction in left-periphery marker usage after controlling for utterance length, speaker identity, and scripted/improvised condition.

5. Under strictly causal (past-only) constraints, our simple models achieve strong performance on both 4-way and 6-way classification, illustrating that competitive accuracy is attainable without access to future turns—a practical consideration for real-time deployment.

## 2 Related Work

### 2.1 Emotion Recognition in Conversation

Early ERC methods focused on capturing conversational dynamics through recurrent architectures. DialogueRNN (Majumder et al., 2019) models speaker states across turns, achieving 76.2% on IEMOCAP 4-way classification. COSMIC (Ghosal et al., 2020) incorporates commonsense knowledge for context enhancement (77.4%). Recent work has pursued increasingly complex architectures, including graph-based models (Ghosal et al., 2019), hierarchical attention (Ma et al., 2022), and multimodal transformers (Hu et al., 2022).

A central axis of variation across ERC systems is how conversational context is operationalized. Several approaches adopt heuristic context windows by concatenating neighboring utterances as raw text; for example, UniMSE concatenates the current utterance with a symmetric window of two preceding and two following turns, which implicitly relies on future context (Hu et al., 2022). Other models assume a fixed number of preceding turns as the local interaction neighborhood, treating context length as a default hyperparameter rather than an empirically justified operating point. For transformer baselines, long contexts are often handled by concatenation with truncation: when the input exceeds the model's length limit, remote utterances are discarded, imposing an implicit cutoff on usable history (e.g., Shen et al. (2021)).

Among high-performing text-only methods, HCAM (Dutta & Ganapathy, 2024) explicitly introduces inter-utterance context via a bidirectional GRU (Bi-GRU) with self-attention, aiming to leverage information from the full conversation. Their formulation uses Bi-GRU to incorporate contextual signals and adds self-attention to better handle long-range dependencies across dialogues with many utterances. While effective, such bidirectional context access is not available in real-time settings where future turns are unknown.

In contrast, we focus on strictly causal (past-only) modeling and explicitly characterize the performance–context trade-off by sweeping the number of preceding turns and quantifying saturation behavior. This complements prior work by replacing heuristic context choices (fixed windows or implicit truncation) with a systematic analysis of how much past text is sufficient, and whether context requirements differ across emotions.

### 2.2 Discourse Markers and Affective Lexicons

Linguistic theory has long recognized discourse markers as signals of speaker stance and emotion (Schiffrin, 1987). Fraser (1999) classified pragmatic markers by function, while Beeching & Detges (2014) demonstrated their tendency to cluster at utterance peripheries. The left periphery often hosts markers expressing speaker stance or discourse management (e.g., *well* signaling hesitation, *oh* marking surprise or realization, and *actually* introducing contrast or correction), while right-peripheral markers can be more hearer-oriented (Traugott, 2010; Beeching & Detges, 2014).

Despite extensive theoretical study, computational ERC models have rarely examined whether such positional patterns vary systematically by emotion. We provide a large-scale corpus analysis of discourse-marker positioning in IEMOCAP and evaluate its association with emotion labels, complementing prior linguistic accounts of peripheral distributions.

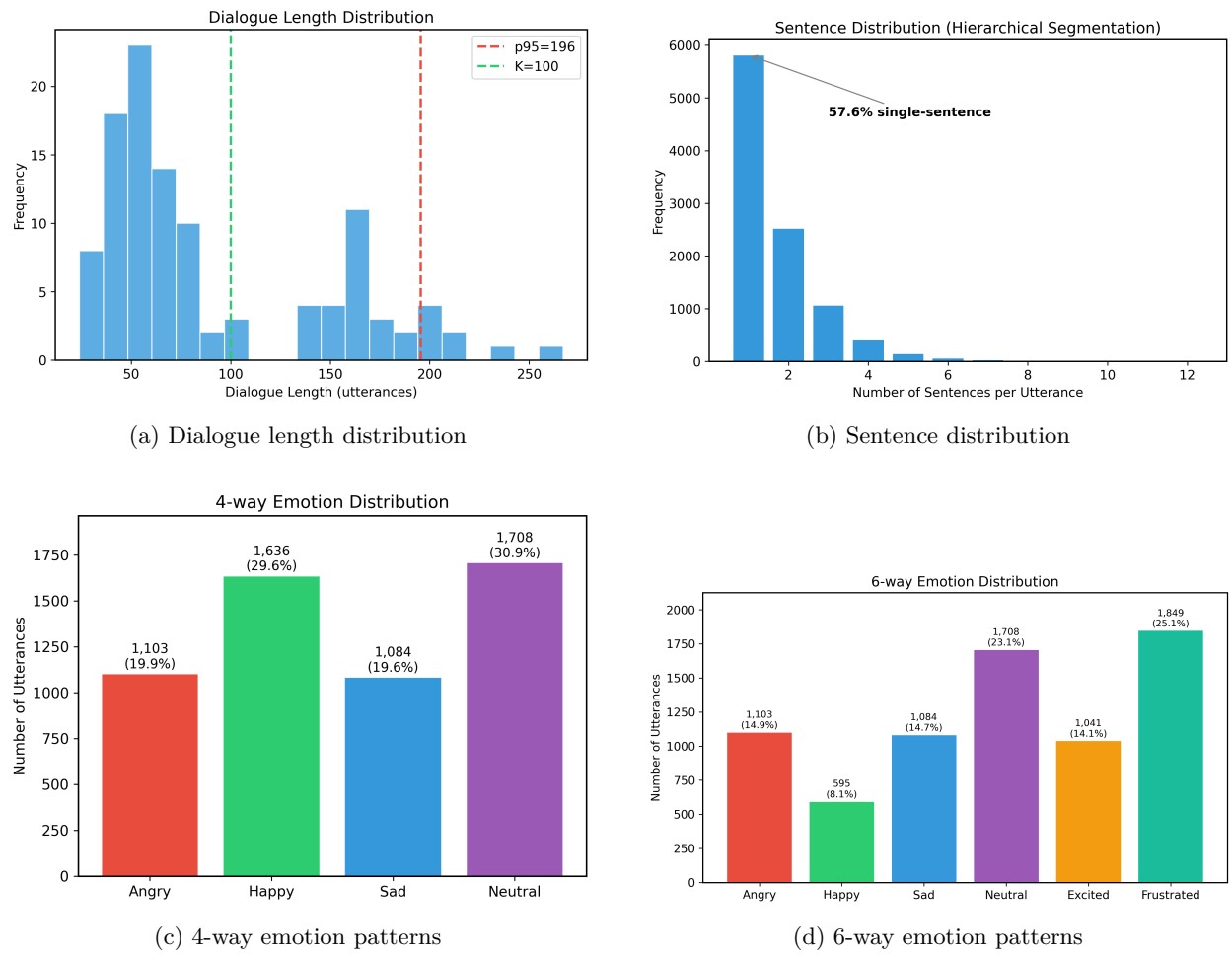

Figure 1: IEMOCAP dataset characteristics.

Affective lexicons provide complementary emotional knowledge. While early resources such as WordNet-Affect (Strapparava, 2004) offered categorical labels, SenticNet (Cambria et al., 2024) maps concepts onto psychological dimensions. Knowledge-enhanced models (Zhong et al., 2020; Tu et al., 2022b) report gains, but whether lexicons add unique information beyond contextual encoders remains unclear—a question we examine empirically under a simple integration scheme.

## 3 Methodology

### 3.1 Task and Dataset

We conduct experiments on the IEMOCAP dataset (Busso et al., 2008), which contains approximately 12 hours of audiovisual data from 10 actors performing scripted and improvised dyadic conversations across 110 dialogue sessions. The dataset exhibits substantial variability in dialogue length (Figure 1a), with sessions ranging from 24 to 267 utterances (mean: 91.7, median: 67.5). Approximately 70% of dialogues contain fewer than 100 utterances, while the 95th percentile reaches 196 utterances, highlighting the long-tail distribution inherent in conversational data.

**Turn-level context length and $K$-sweep protocol.** We define the context length $K$ as the number of *preceding turns* included as strictly past-only conversational history for a target utterance. Because there is no established stopping rule for how much prior dialogue is sufficient in ERC, we perform a bottom-up sweep

$K \in \{0, 1, \ldots, K_{\max}\}$ to characterize the performance–context trade-off and identify saturation behavior. We set $K_{\max} = 200$ based on the dialogue-length distribution (covering the long tail while keeping the sweep computationally tractable).[1] Crucially, the sweep is used strictly for analysis (saturation characterization) and not for test-set model selection; we do not choose $K$ to maximize test performance.

When a target utterance has fewer than $K$ preceding turns (e.g., early in a dialogue), we use all available history (a variable-length prefix). For turn-level models, sequences are left-padded for batching and padded positions are masked so they do not contribute to contextual aggregation.

For a fair comparison with state-of-the-art models (Dutta & Ganapathy, 2024), we follow the standard speaker-disjoint split strategy: Sessions 2–4 serve as training data (6,072 utterances from 68 dialogues), Session 1 as validation (1,819 utterances from 20 dialogues), and Session 5 as test set (2,196 utterances from 22 dialogues). This split ensures speaker independence between training and evaluation, with each dialogue containing an average of 89–100 utterances across splits. The sentence-level structure within utterances (Figure 1b) shows that most utterances contain 2–5 sentences, motivating our comparison of flat versus hierarchical encoding strategies. Sentence segmentation for hierarchical encoding is performed with spaCy's sentencizer after lowercasing and standard token normalization.[2]

We evaluate on two emotion taxonomies commonly used in the literature. The 4-way classification task considers angry (1,103 utterances, 19.9%), happy (1,636, 29.6%), sad (1,084, 19.6%), and neutral (1,708, 30.9%) emotions, where the excited category is merged into happy (Figure 1c). The 6-way task extends this to include excited (1,041, 14.1%) and frustrated (1,849, 25.1%) as separate categories (Figure 1d). We use weighted F1-score as our primary metric to account for class imbalance inherent in conversational emotion data, and we also report class-wise F1 scores and confusion matrices for error analysis.

### 3.2 Discourse Marker Analysis

Beyond recognition accuracy, we conduct a linguistic analysis of discourse markers (DMs) to examine emotion-specific pragmatic patterns in conversational discourse. DMs are lexical expressions that signal relationships between discourse segments rather than contributing to propositional content (Schiffrin, 1987; Fraser, 1999). Drawing from established taxonomies (Schiffrin, 1987; Fraser, 1999; Traugott, 2010; Beeching & Detges, 2014), we identify 20 markers occurring in IEMOCAP, including turn-management markers (*well, oh*), connectives (*and, but, so*), and stance markers (*I think, I guess, maybe, you know, I mean*). See Appendix A for the full inventory and frequency distribution.

For each marker occurrence, we record its relative position within the utterance (normalized to $[0, 1]$), its periphery classification, and the emotion label of the containing utterance. Following standard discourse accounts of utterance peripheries (Beeching & Detges, 2014), we operationally define the left periphery (LP) as position $< 0.15$, the right periphery (RP) as position $> 0.85$, and medial otherwise. For consistency with our pooling notation, *wmean_pos_rev* emphasizes utterance-initial (left-peripheral) tokens, whereas *wmean_pos* emphasizes utterance-final (right-peripheral) tokens; mean pooling treats all positions uniformly.

To test for emotion-specific positional patterns, we employ (i) ANOVA to compare mean positions across emotions, (ii) $\chi^2$ tests with Cramér's $V$ to assess association between periphery categories and emotions, and (iii) mixed-effects models with dialogue as a random intercept to control for dialogue-level variation (e.g., $\text{pos} \sim \text{emotion} + (1|\text{dialogue})$). We apply post-hoc pairwise comparisons with Bonferroni correction where appropriate. For robustness, we additionally estimate confound-control models that include utterance length, speaker identity, and scripted/improvised condition; these analyses are reported in Section 4.

### 3.3 Model Architecture

To investigate how emotional information is encoded and propagated in dialogue, we develop two encoder variants: a flat encoder and a hierarchical encoder. The flat encoder processes each utterance as a single

---

[1]The maximum dialogue length in IEMOCAP is 267 turns; our choice of $K_{\max} = 200$ is driven by the empirical distribution (Figure 1a), where 200 exceeds the 95th percentile, and is used for analysis rather than model selection.

[2]We follow common ERC preprocessing practice (lowercasing and tokenization) and use a deterministic sentence boundary detector to ensure reproducibility.

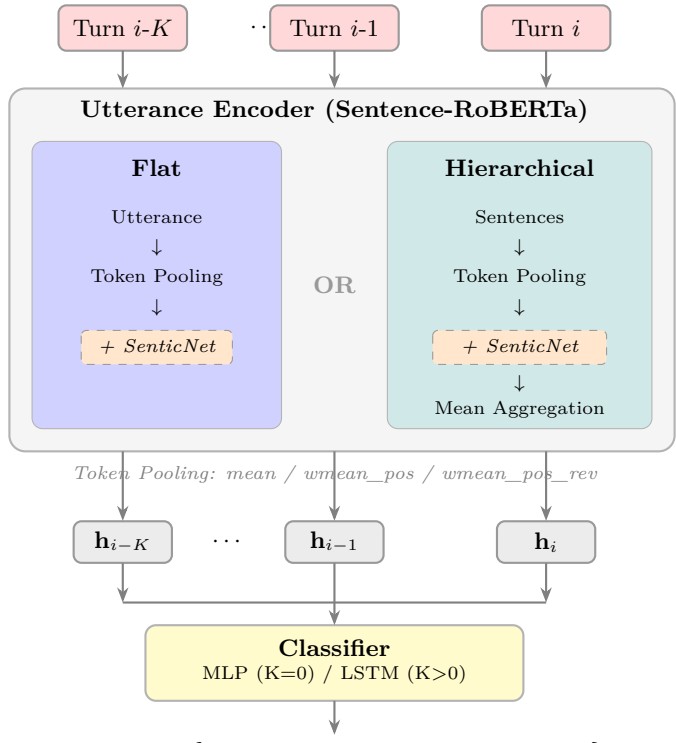

Figure 2: Model architecture. Each turn is encoded independently via Sentence-RoBERTa using either flat (whole utterance) or hierarchical (sentence-level) encoding. SenticNet features are optionally fused (dashed boxes). For classification, we use MLP when K=0 (no context) or unidirectional LSTM when K>0 (with preceding turns as context).

sequence, while the hierarchical encoder first encodes individual sentences within an utterance, then aggregates them to form the utterance representation. Figure 2 illustrates our architecture.

**Utterance Encoder.** We compare (1) *flat* encoding, which treats each utterance as a single sequence and extracts a pooled representation, and (2) *hierarchical* encoding, which encodes each sentence and then aggregates sentence representations into an utterance vector. We evaluate three pre-trained encoders: BERT-base-uncased (Devlin et al., 2019), RoBERTa-base (Liu et al., 2019), and Sentence-RoBERTa (NLI-RoBERTa-base-v2) (Reimers & Gurevych, 2019). Unless stated otherwise, encoders are used as fixed feature extractors and utterance embeddings are precomputed to isolate the effects of contextual modeling and pooling choices.[3]

We compare two layer strategies: (i) *avg_last4*, averaging the last four transformer layers, and (ii) *last*, using only the final layer output. We use a maximum input length of 128 subword tokens for the utterance encoder; longer utterances are truncated from the left/right as in standard transformer preprocessing.

**Context construction and label availability.** During embedding generation, we include all turns in each dialogue in chronological order, including utterances without target emotion labels or outside the target label set. These turns may appear in the contextual history, but only target-labeled utterances contribute to the training loss and evaluation metrics. This allows long-range history to be represented without altering the supervised label space.

---

[3]This design avoids confounding the analysis with end-to-end fine-tuning instability across seeds.

**Classifier.** For utterance-level classification ($K{=}0$), the utterance representation is passed through a two-layer MLP with ReLU activation and dropout. When incorporating turn-level context ($K{>}0$), we process the sequence of utterance representations via a single-layer unidirectional LSTM and use the final hidden state for classification.

**Lexical Feature Integration.** To examine whether external affective knowledge benefits emotion recognition, we integrate SenticNet 8 (Cambria et al., 2024), which provides four-dimensional affective ratings (pleasantness, attention, sensitivity, aptitude) for words and phrases. We lowercase and tokenize each utterance, match SenticNet entries at the token level (and, where available, multiword expressions), and aggregate matched vectors via mean pooling; unmatched tokens contribute zeros. The resulting 4D vector is concatenated with the encoder representation.

### 3.4 Training and Implementation Details

Our classifier consists of a two-layer MLP (for $K{=}0$) or a single-layer unidirectional LSTM (for $K{>}0$). We use fixed hyperparameters across all experiments: learning rate $1e{-}3$, hidden dimension 256, dropout 0.3, and batch size 64. Hyperparameters were selected once on the validation set and then held fixed across all ablations to ensure comparability.

Training employs Adam with early stopping on validation weighted F1 (patience 60 for utterance-level and 20 for turn-level experiments).[4] All experiments use 10 random seeds $\{42, 43, \ldots, 51\}$ with results reported as mean $\pm$ standard deviation. Statistical comparisons across configurations use paired $t$-tests over seeds with Bonferroni correction where applicable. Implementation uses PyTorch 1.13, and experiments were conducted on NVIDIA A100 GPUs via Saturn Cloud.

### 3.5 Encoder Selection

We compared three pre-trained text encoders at utterance level ($K{=}0$) on IEMOCAP 4-way classification (Table 1). Sentence-RoBERTa yields the highest mean weighted F1 (65.29%), which is plausibly attributable to its NLI fine-tuning that produces robust sentence-level semantics. We emphasize, however, that this ranking may be dataset-dependent: IEMOCAP consists of relatively well-structured, acted dyadic dialogues, and noisier, multi-party corpora (e.g., MELD) may favor different encoders. To isolate architectural effects under a fixed encoder, all subsequent experiments use Sentence-RoBERTa unless otherwise noted.

Table 1: Encoder comparison on 4-way classification (K=0, 10 seeds).

| Encoder | WF1 (%) | Std | Min | Max | 95% CI |
|---|---|---|---|---|---|
| BERT-base | 63.99 | 0.85 | 62.62 | 65.18 | [63.38, 64.59] |
| RoBERTa-base | 65.01 | 0.80 | 63.90 | 66.31 | [64.44, 65.58] |
| Sentence-RoBERTa | **65.29** | 1.17 | 64.10 | 67.31 | [64.45, 66.12] |

### 3.6 Main Results

We report two complementary views of performance. First, for a fair comparison across design choices, we evaluate each configuration at fixed context sizes $K$ (Table 2), where $K$ denotes the number of strictly preceding turns (past-only context). This controlled evaluation shows that conversational context dominates performance: most of the total gain is already realized with a short history (e.g., $K{=}10$), and performance changes only marginally beyond $K{=}30$–$50$ depending on the task.

Within the same $K$, neither hierarchical encoding nor position-weighted pooling shows a consistent advantage. In particular, FLAT vs. HIER differences are not statistically significant in the fixed-$K$ setting (paired tests across seeds; $p{>}0.9$), and pooling choices do not yield reliable improvements (all comparisons $p{>}0.08$).

---

[4]We use validation weighted F1 to align the early-stopping criterion with the primary evaluation metric.

Table 2: Fixed-$K$ evaluation on IEMOCAP with Sentence-RoBERTa (10 seeds; mean±std where reported). $K$ is the number of preceding turns (past-only context). For a small number of $K$=100/$K$=200 cells the std was not recovered from the original sweep logs at the time of writing (point estimate only). These are clearly marked by the absence of a ± value and lie within the post-saturation plateau. The 6-way HIER `wmean_pos_rev` value at $K$=200 is computed over $n$=9 seeds.

| Task | Type | Pooling | $K$=0 | $K$=10 | $K$=30 | $K$=50 |
|---|---|---|---|---|---|---|
| 4-way | FLAT | mean | 64.94±0.77 | 79.44±1.10 | 80.56±1.15 | 80.66±1.15 |
| 4-way | FLAT | wmean_pos | 64.80±0.87 | 79.21±0.98 | 80.63±0.84 | 80.28±0.93 |
| 4-way | HIER | mean | 64.52±1.01 | 78.39±1.29 | 80.64±1.88 | 78.99±2.70 |
| 4-way | HIER | wmean_pos | 64.03±1.15 | 77.94±1.14 | 79.17±1.04 | 80.22±1.09 |

| Task | Type | Pooling | $K$=0 | $K$=10 | $K$=30 | $K$=50 |
|---|---|---|---|---|---|---|
| 6-way | FLAT | mean | 52.35±1.36 | 62.80±2.32 | 64.77±1.26 | 64.58±1.07 |
| 6-way | FLAT | wmean_pos | 52.44±0.78 | 64.25±1.25 | 64.26±1.66 | 64.38±1.40 |
| 6-way | HIER | mean | 51.54±1.04 | 63.24±1.58 | 64.74±1.59 | 63.85±1.86 |

| Task | Type | $K$=100 | $K$=200 |
|---|---|---|---|
| 4-way | FLAT (mean) | 80.60±1.56 | 79.59±1.64 |
| 4-way | FLAT (wmean_pos) | 79.52±1.76 | 80.13±1.01 |
| 4-way | HIER (mean) | 79.43±2.15 | 79.08 |
| 4-way | HIER (wmean_pos) | 78.95±1.76 | 77.81 |
| 6-way | FLAT (mean) | 64.11±1.31 | 63.72±1.58 |
| 6-way | FLAT (wmean_pos) | 65.40±1.08 | 64.44±1.35 |
| 6-way | HIER (mean) | 63.72±1.14 | 62.62±1.50 |
| 6-way | HIER (wmean_pos_rev) | 62.49±2.37 | 63.74±1.53 |

These results suggest that once modest conversational history is provided, architectural sophistication at the utterance level contributes little relative to context.

Second, to characterize the performance–context trade-off and identify saturation behavior, we also conduct a full $K$-sweep analysis (Section 3.7). For completeness, we additionally report the best-performing configurations observed within the fixed-$K$ evaluations across the sweep range (Table 2); importantly, these are used for analysis rather than test-set model selection.

## 3.7 Emotion-Specific Context Effects

While prior work on variable-length context focuses on *how* to adaptively select context windows through speaker-aware modules (Zhang et al., 2023), we take a complementary approach: we systematically investigate *what* patterns emerge when varying context length and *whether* different emotions exhibit distinct context requirements. We present this analysis in Figure 3, framing the per-emotion performance–context trade-off through saturation behavior rather than the location of in-sweep maxima.

To analyze emotion-specific effects, we compute per-class F1 scores at each context length from $K$=0 (utterance only) to $K$=200 (preceding turns). For each emotion, we report the context length at which F1 attains its maximum within the sweep range, and quantify improvement relative to $K$=0. We also determine the saturation point, defined as the minimum $K$ at which 90% of the maximum improvement (within the sweep range) is achieved. All analyses are conducted across 10 random seeds.

Our first finding is that performance saturates rapidly in aggregate: a short history (e.g., $K$=10) already recovers most of the eventual gain in weighted F1, and improvements beyond $K$=30–50 are modest for the average utterance (Table 2). This suggests that immediate preceding context carries the majority of predictive information for ERC under strictly causal access. Importantly, per-emotion F1 at $K$=0 is already non-trivial (between 0.61 and 0.70 in the 4-way setting; Figure 3a), indicating that utterance-level lexical and pragmatic

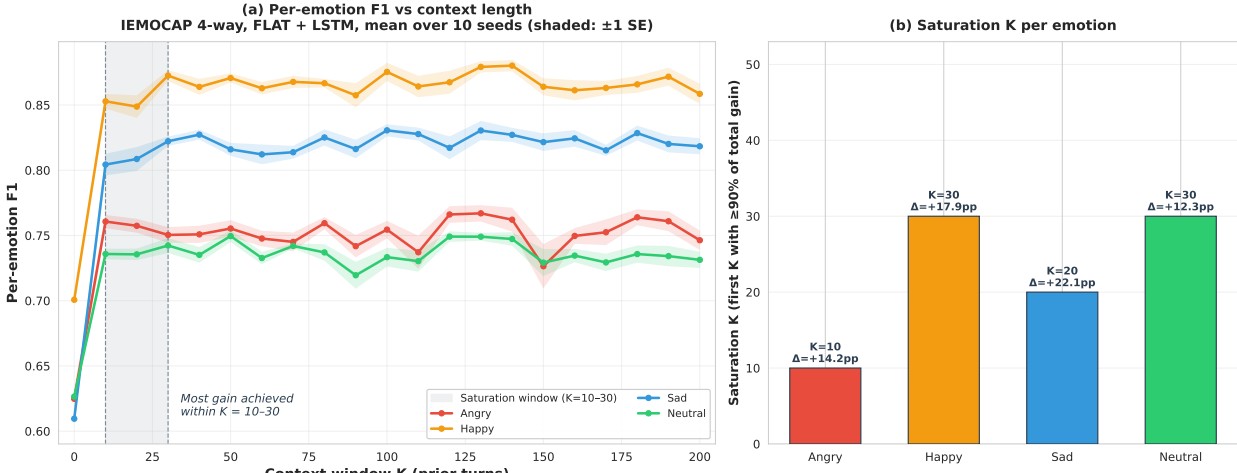

Figure 3: Per-emotion F1 versus context length on IEMOCAP (4-way, frozen Sentence-RoBERTa + LSTM, mean over 10 seeds, $\pm 1$ SE shaded). **(a)** The bulk of gain over $K{=}0$ is realised within the shaded $K{=}10$–$30$ window for all four emotions; fluctuations beyond this window remain within seed-level noise. **(b)** *Saturation $K$* (bars), the smallest $K$ at which 90% of total F1 improvement over $K{=}0$ is achieved, clusters in $[10, 30]$ for every emotion. Per-bar annotations report $\Delta F1$, the total improvement over $K{=}0$ realised within the sweep. Late numerical maxima of the per-emotion curves fall within the post-saturation plateau and are discussed in the main text; we do not draw them here to avoid implying that very long histories are required.

cues alone provide a meaningful emotional signal. Conversational context thus acts as a complement to, rather than a prerequisite for, emotion recognition, with the degree of benefit varying across emotions.

Our second finding reveals a dissociation between saturation timing and improvement magnitude at the emotion level. While emotions reach saturation at broadly similar context lengths (Kruskal–Wallis $H = 0.52$, $p = 0.91$), the magnitude of improvement differs substantially (one-way ANOVA $F = 136.80$, $p < 0.0001$). Sad benefits the most from context ($+22\%$p), whereas Angry benefits the least ($+8$–$9\%$p), and this ordering remains consistent across 4-way and 6-way settings. Concretely, the per-emotion saturation $K$ values cluster in $[10, 30]$ for all four 4-way emotions (Figure 3b). Beyond $K{=}30$, additional turns produce fluctuations on the order of $\pm 1\%$p but do not yield reliable improvement; consequently the apparent "peak $K$" for each emotion (e.g., $K^*{=}130$ for Angry, $K^*{=}140$ for Happy in the 4-way sweep) falls within this noisy plateau and should not be interpreted as a requirement for one hundred or more turns of history. We also note that this analysis treats conversational history as a sequence rather than an undifferentiated bag of turns: the LSTM's recurrent hidden state implicitly tracks emotional trajectory through its gating mechanism, selectively retaining or down-weighting earlier turns based on their relevance to the current prediction.

Taken together, these results reframe context as a complement rather than a prerequisite: a $K{=}0$ baseline already attains 0.61–0.70 F1 per emotion, $K{=}10$–$30$ captures the practically meaningful contextual contribution, and the long-tail "peak" $K$ values seen at $K{>}100$ reflect within-plateau noise rather than a genuine requirement for very long conversational histories. We turn next to a cross-dataset validation that directly addresses the dataset-specific scope of these claims.

### 3.8 Cross-Dataset Validation on MELD

To assess whether our core findings reflect general properties of conversational emotion recognition or characteristics specific to IEMOCAP, we replicate the central experiments on MELD (Poria et al., 2018), a 7-way emotion classification dataset derived from *Friends*. MELD differs from IEMOCAP along several structurally relevant dimensions: multi-party TV-show dialogue rather than controlled dyadic IEMOCAP sessions, and an average dialogue length of 9.6 turns versus 89.3 in IEMOCAP. These differences let us

Table 3: Cross-dataset validation on MELD (7-way, frozen Sentence-RoBERTa, 10 seeds; mean $\pm$ std weighted F1). Protocol matches the IEMOCAP experiments exactly.

| Encoding | $K{=}0$ | $K{=}5$ | $K{=}10$ |
|---|---|---|---|
| FLAT | $59.22 \pm 0.51$ | $58.90 \pm 0.63$ | $59.07 \pm 0.73$ |
| HIER | $61.48 \pm 0.66$ | $61.02 \pm 0.72$ | $60.91 \pm 0.75$ |

probe whether the saturation behavior and hierarchical-encoding patterns are dataset artifacts or general regularities.

We use the identical experimental protocol as in our IEMOCAP experiments: frozen Sentence-RoBERTa (`avg_last4`, mean pooling), MLP head for $K{=}0$, unidirectional LSTM for $K{>}0$, 10 random seeds, and the same hyperparameters (hidden size 256, dropout 0.3, learning rate $10^{-3}$, batch size 64, patience 10). Class weights and additional fusion components are deliberately omitted to keep the comparison architecturally aligned with the IEMOCAP analysis. Table 3 reports weighted F1 across the three relevant context lengths.

The MELD results support the saturation-based interpretation and refine the scope of our IEMOCAP findings. **(1) Hierarchical encoding helps when context is absent.** HIER exceeds FLAT by $+2.26$ p.p. at $K{=}0$ (paired test across seeds, $p{<}0.001$), confirming that intra-utterance sentence structure is useful in utterance-only settings and that this benefit is more pronounced in noisier multi-party dialogue. **(2) Context saturates immediately on MELD.** Both encodings show no improvement (and in fact a slight decrease) when context is added at $K{=}5$ or $K{=}10$. This is precisely what the saturation framing of Section 3.7 predicts: with an average of 9.6 utterances per dialogue, the contextual signal available within a short window is already exhausted, and the per-emotion saturation $K$ effectively collapses toward $K{=}0$. The IEMOCAP saturation window of $[10, 30]$ thus reflects the longer scale of meaningful conversational dynamics in that corpus, not a universal requirement. **(3) Emotion-specific context dependencies persist, but their class-level expression is corpus-specific.** Per-class analysis shows that anger improves the most with context on MELD ($+2.8$ p.p. from $K{=}0$ to $K{=}10$ with HIER), while joy, neutral, and surprise show no reliable change. The set of emotions that benefit from context differs from IEMOCAP (where sadness benefits most), suggesting that the identity of the most context-dependent class is corpus-specific.

Together, these results support the cross-dataset relevance of the saturation framing while clarifying the scope of the hierarchical-encoding result. Hierarchical encoding shows its clearest advantage at $K{=}0$ on MELD, whereas rapid saturation appears in both corpora but at different operating points. We therefore scope our context-length claims to the dialogue regime under study. Practical saturation occurs within the first few turns to a few tens of turns of relevant conversational history, with the exact value tracking the typical dialogue length of the corpus.

## 3.9 Ablation Studies

We conducted ablations over (i) pooling strategy, (ii) external lexical knowledge, and (iii) layer aggregation, using the same training protocol and 10-seed evaluation as in the main experiments.

**Pooling strategy.** We compared mean pooling, position-weighted pooling emphasizing utterance-final tokens (`wmean_pos`), and position-weighted pooling emphasizing utterance-initial tokens (`wmean_pos_rev`). Across seeds, no pooling method was consistently superior (Friedman test, $p > .08$). In terms of best-performing configurations, position-weighted pooling more often appeared among the top runs at utterance-level ($K{=}0$), whereas mean pooling was more frequently competitive when conversational context was included ($K{>}0$); however, these differences were not statistically reliable.

**External lexical knowledge.** We augmented utterance representations with SenticNet 8 (Cambria et al., 2024), which provides four affective dimensions (pleasantness, attention, sensitivity, aptitude). Following common practice, we tested a simple integration scheme: per-token SenticNet vectors are mean-pooled into a 4-dimensional utterance descriptor and concatenated with the encoder representation. This augmentation

Table 4: Transformer vs. LSTM aggregator comparison. Values are weighted F1. MELD rows are formal like-for-like 10-seed comparisons. IEMOCAP rows are auxiliary diagnostics: LSTM values are 10-seed HIER *wmean_pos* K-sweep means, while Transformer values marked with $^\dagger$ are single-seed runs under the same HIER *wmean_pos* configuration.

| Dataset | Encoding | $K$ | LSTM | Transformer | $\Delta$ |
|---|---|---|---|---|---|
| IEMOCAP 4-way | HIER | 10 | $77.75 \pm 1.17$ | $72.77^\dagger$ | $-5.0$ |
| | HIER | 20 | $78.46 \pm 0.90$ | $71.28^\dagger$ | $-7.2$ |
| | HIER | 30 | $79.21 \pm 0.94$ | $70.51^\dagger$ | $-8.7$ |
| MELD 7-way | FLAT | 5 | $58.90 \pm 0.63$ | $58.90 \pm 0.66$ | $0.00$ |
| | FLAT | 10 | $59.07 \pm 0.73$ | $58.86 \pm 0.56$ | $-0.21$ |
| | HIER | 5 | $61.02 \pm 0.72$ | $60.39 \pm 0.76$ | $-0.63$ |
| | HIER | 10 | $60.91 \pm 0.75$ | $60.30 \pm 0.81$ | $-0.61$ |

$^\dagger$Single-seed diagnostic Transformer run, seed 42; per-$K$ Transformer seed variance is not available. For IEMOCAP, $\Delta$ is descriptive only.

did not improve performance and in some settings slightly reduced it (e.g., 4-way: $\approx -0.94$ F1 points; 6-way: near-zero change), suggesting that the pretrained encoder representations already capture the affective information that SenticNet provides under this simple integration scheme. Full results across all configurations are reported in Appendix C. We emphasize that this null result is specific to concatenation-based fusion; more sophisticated integration mechanisms (e.g., graph-based or attention-driven approaches as in Zhong et al., 2020; Tu et al., 2022b) may yield different outcomes and remain an open question for future work.

**Layer aggregation.** We compared averaging the last four transformer layers (`avg_last4`) against using only the final layer (`last`). We observed no significant differences (paired *t*-tests; minimum $p = .244$ across comparisons).

**Transformer vs. LSTM aggregator.** To probe whether rapid context saturation could be an artifact of the unidirectional LSTM aggregator, we compare the LSTM with a 2-layer Transformer encoder (4 attention heads, mean pooling over positions, sinusoidal positional encoding). Results are reported in Table 4.

Our formal comparison is on MELD, where both aggregators are trained for 10 random seeds with identical sentence pooling and otherwise matched hyperparameters (frozen Sentence-RoBERTa encoder, same hidden size, optimiser, and patience). The Transformer performs equal to or slightly worse than the LSTM at every condition tested: the LSTM is ahead by at most 0.63 p.p. on HIER and is indistinguishable from the Transformer on FLAT, and no condition shows a Transformer advantage. We additionally report an IEMOCAP 4-way diagnostic comparison under the same HIER *wmean_pos* sentence pooling, with the LSTM aggregated over 10 seeds and the Transformer ($\dagger$ in Table 4) reported from the only single-seed (seed 42) K-sweep available at submission time. With that single-seed caveat on the Transformer side, the IEMOCAP diagnostic runs likewise do not suggest an obvious Transformer gain.

Taken together, the 10-seed MELD comparison and the matched-pooling IEMOCAP diagnostic check suggest that the $K$=10–30 saturation observed in Section 3.7 is not solely attributable to the limited capacity of the LSTM aggregator under the frozen-encoder context-aggregation setting tested here. A multi-seed IEMOCAP Transformer K-sweep would further sharpen the comparison and is left to future work.

### 3.10 Comparison with Prior Work

We situate our results against prior text-only ERC methods on IEMOCAP. A key axis of comparison is temporal access: bidirectional models can use both past and future utterances, whereas past-only (causal) models are restricted to preceding context and are therefore deployable in real-time settings. Importantly, prior results are reported under heterogeneous protocols (e.g., context windows, architectures, and often

single-run reporting), so the tables below provide a reference comparison rather than a strict head-to-head evaluation.

**4-way classification.** Table 5 shows that our past-only model is competitive with, and in some cases exceeds, reported performance of representative text-only baselines, including several bidirectional systems. We report the mean across 10 seeds for statistical robustness, whereas most prior work reports a single number.

Table 5: IEMOCAP 4-way (text-only). Our result is mean over 10 seeds; prior work numbers are reported in their respective papers. †Text-only ablations reported in multimodal papers.

| Method | Context | WF1 (%) |
|---|---|---|
| **Ours (mean over 10 seeds)** | Past-only | **82.69** |
| HFFN† (Mai et al., 2019) | Bidirectional | 81.54 |
| HCAM (Dutta & Ganapathy, 2024) | Bidirectional | 81.4 |
| CHFusion† (Majumder et al., 2018) | Bidirectional | 73.6 |

**6-way classification.** On the 6-way task (Table 6), our past-only model achieves 67.07% weighted F1 (mean over 10 seeds), which is comparable to strong reported baselines and exceeds several bidirectional systems. We stress that differences should be interpreted cautiously due to protocol mismatch and the prevalence of single-run reporting in prior work.

Table 6: IEMOCAP 6-way (text-only, excluding LLM-based approaches). Our result is mean over 10 seeds; prior work numbers are reported. Bold indicates best reported past-only result among non-LLM baselines.

| Method | Context | WF1 (%) |
|---|---|---|
| EmoCaps (Li et al., 2022c) | Bidirectional | 69.49 |
| DAG-ERC (Shen et al., 2021) | Past-only | **68.03** |
| **Ours (mean over 10 seeds)** | Past-only | 67.07 |
| SKAIG (Li et al., 2021) | Bidirectional | 66.96 |
| DialogueCRN (Hu et al., 2021) | Bidirectional | 66.20 |
| CoG-BART (Li et al., 2022a) | Bidirectional | 66.18 |
| BiERU (Li et al., 2022b) | Bidirectional | 64.59 |
| HCAM (Dutta & Ganapathy, 2024) | Bidirectional | 64.4 |
| DialogueGCN (Ghosal et al., 2019) | Bidirectional | 64.18 |

## 4 Discussion

Our study delivers strong performance under a strictly causal (past-only) setting (82.69% for 4-way; 67.07% for 6-way) and yields several interpretable patterns. Across analyses, conversational context provides the dominant gain, emotion classes differ markedly in their reliance on context, and external lexicons (SenticNet) do not improve performance. In addition, our corpus-based analysis of 5,286 discourse-marker (DM) occurrences reveals small but reliable emotion-specific positional tendencies. Below, we discuss four implications.

### 4.1 Conversational Context Reduces the Marginal Utility of Hierarchical Structure

We observe an interaction between encoding strategy and context availability. At $K=0$, hierarchical and flat encodings are not statistically separable on IEMOCAP (paired tests across seeds, $p>0.9$; cf. Section 3.6 and Table 2, $K=0$ columns), whereas a clearer hierarchical advantage emerges on MELD, where HIER exceeds FLAT by +2.26 p.p. at $K=0$ across 10 seeds (Section 3.8). This cross-dataset pattern is consistent with the intuition that intra-utterance sentence structure can provide additional cues when the model cannot consult

surrounding turns, and that this benefit is more pronounced in noisier, multi-party dialogue where individual utterance signals are weaker.

However, once past conversational context is incorporated, this intra-utterance structural advantage diminishes on both datasets: flat and hierarchical variants become closely matched, and the best-performing contextual models are not reliably separated by the choice of utterance encoder. This suggests that turn-level context can partly substitute for fine-grained intra-utterance structure, because emotional evidence is often distributed across adjacent turns (e.g., responses, clarifications, escalation/de-escalation). Practically, this supports using simpler flat encoders in causal, real-time deployments when sufficient past context is available.

## 4.2 Do Utterance Peripheries Carry Emotion-Relevant Signal?

Pooling variants did not yield statistically significant differences in our multi-seed comparisons (Friedman $p > .08$), yet we consistently observed that position-weighted pooling is competitive at $K=0$, while simple mean pooling is sufficient once context is introduced. We interpret this as a weak but suggestive pattern rather than a definitive effect.

This interpretation is compatible with our discourse-marker analysis. We find a statistically reliable but small association between emotion and DM periphery category ($\chi^2$, $p < .0001$; Cramér's $V = 0.062$, which by conventional benchmarks is a negligible-to-small effect). We therefore interpret this association as a probabilistic tendency in marker positioning. Its main value lies in linking the recognition results to a linguistically grounded discourse pattern. We do not treat discourse-marker position as a strong standalone predictive feature. Notably, Sad utterances show reduced left-periphery usage (21.9%) compared to Neutral (31.7%), Happy (29.7%), and Angry (28.2%), with post-hoc tests indicating that Sad differs from the other emotions (Bonferroni-corrected, all $p < .01$). One plausible account is that left-periphery markers (e.g., *well, oh*) often participate in turn-management and stance negotiation. Their reduced use may reflect more muted pragmatic signaling in Sad speech.

To rule out alternative explanations for these positional differences, we re-estimated the emotion–position association using mixed-effects logistic regression with progressive addition of covariates. The five-stage analysis (full model on $N$=5,286 marker occurrences) yields the following picture. The baseline emotion-only model recovers a positive Sad coefficient ($+0.051$, $p < 0.001$). Adding log-transformed utterance length as a fixed effect renders the Angry and Happy effects non-significant ($p > 0.4$) but leaves Sad significant (coefficient $+0.035$, $p = 0.011$). Adding speaker as a random effect contributes negligible variance and does not alter the Sad result. Adding a scripted-versus-improvised indicator has only a marginal effect ($p \approx 0.025$) on the covariate itself and leaves Sad significant ($+0.039$, $p = 0.005$). A binary left-periphery model in the full specification yields Sad coefficient $-0.104$ ($p < 0.001$). The reduced left-periphery use in Sad survives simultaneous control for utterance length, speaker identity, and elicitation condition, whereas the apparent Angry and Happy effects from the marginal $\chi^2$ are largely attributable to utterance length. We interpret the discourse-marker result primarily through the *Sad* category. In this corpus, Sad utterances show reliably reduced left-periphery marker use after these controls. This pattern suggests that sadness may be associated with weaker overt discourse-management signaling in IEMOCAP.

## 4.3 Why Does Sadness Benefit Most from Context?

Emotion-specific analysis shows that Sad gains the most from added context ($+22.31$%p), whereas Angry gains the least ($+8.34$%p). A pure arousal-based account is insufficient, because Happy (often high-arousal) also benefits substantially from context ($+15.82$%p). Instead, our results suggest that what matters is the availability of explicit lexical/pragmatic cues in the target utterance.

Angry turns often contain salient lexical signals (e.g., emphatic negation, direct accusations, profanity) that are informative even without prior history. By contrast, Sad turns can be lexically understated and pragmatically ambiguous (e.g., *I see*, *yeah*, *I guess*), making their emotional interpretation contingent on the preceding trajectory. This aligns with the DM finding that Sad shows reduced left-periphery marking, potentially reducing overt discourse-management signals and increasing reliance on conversational history

for disambiguation. More broadly, this supports the view that causal ERC should be evaluated not only by overall F1 but also by how different emotions depend on context.

### 4.4 Is the 6-way Taxonomy Well-Identified in Text-Only ERC?

Our confusion patterns raise questions about how well the 6-way categories are separable from text alone. In particular, Happy–Sad confusion is higher than Happy–Excited confusion in our analysis, and Happy absorbs a substantial portion of Excited instances, yielding an asymmetric confusion pattern. This is consistent with the interpretation that acoustic intensity cues—often crucial for distinguishing Excited from Happy—are absent in text-only settings, so the model defaults to the more frequent positive label.

These observations do not imply that the 6-way taxonomy is intrinsically invalid; rather, they suggest that in text-only ERC, some category boundaries may be weakly identified. This motivates alternative formulations, such as (i) hierarchical classification (e.g., valence first, then finer labels), (ii) dimensional prediction (valence/arousal), or (iii) reduced label sets that merge categories that are difficult to separate without prosody.

In the absence of acoustic features, several text-side cues may help disambiguate same-valence high-arousal states such as Excited from their lower-arousal counterparts. Candidate signals include (a) intensifiers and boosters (e.g., *so*, *very*, *really*, *absolutely*) that frequently co-occur with high-arousal expressions; (b) exclamatory punctuation and capitalisation patterns where available in the transcription; (c) lexical repetition and elongation that often accompany excited speech; and (d) the pragmatic affordances captured by the discourse-marker inventory introduced in Section 3.2, whose usage profiles for Happy and Excited are almost identical (Pearson $r = 0.990$, $p < 0.001$). The last observation is itself informative. At the level of discourse-marker categories, Happy and Excited share much of their pragmatic structure. This helps explain why a categorical text-only classifier may conflate them and suggests that future work should consider dimensional formulations, such as valence and arousal, for same-valence emotions that differ primarily in intensity.

## 5 Limitations and Future Directions

While our analyses reveal consistent and interpretable patterns in how causal ERC models leverage context and discourse cues, several limitations should be acknowledged and motivate future extensions.

**Statistical tendencies rather than deterministic rules.** The discourse-marker effects we report are statistically reliable but small in magnitude (e.g., Cramér's $V = 0.062$). Emotional expression is highly variable across speakers, interactional goals, and situations; accordingly, our positional patterns should be interpreted as probabilistic tendencies rather than deterministic linguistic laws. Future work should quantify robustness under domain shift and across interaction types (e.g., multi-party chat in MELD (Poria et al., 2018), written dialogue in DailyDialog, and other conversational corpora), and should report effect sizes alongside significance to avoid overinterpretation.

**Scope of text-only modeling.** Our experiments are restricted to text-based ERC. This captures lexical and discourse-level cues but omits prosodic and visual signals that are often crucial for separating closely related categories (e.g., *excited* vs. *happy*). Extending the same ablation and fixed-context protocol to multimodal encoders would allow precise attribution of what additional information is contributed by acoustics and facial dynamics beyond text.

**Dataset bias, label noise, and generalizability.** IEMOCAP consists of acted English dyadic dialogues, which may differ from spontaneous interaction in turn-taking style, emotional intensity, and marker usage. In addition, categorical emotion labels in conversational corpora are inherently noisy and sometimes underspecified for text-only interpretation. The emotion-specific context patterns we observe (e.g., larger gains for Sad than Angry) may therefore partially reflect dataset- and annotation-specific properties. Cross-dataset replication and multilingual evaluation are essential to determine whether these are general computational regularities or corpus-contingent effects. Our MELD replication (Section 3.8) addresses a first step of this concern by reproducing the saturation behaviour and hierarchical-encoding pattern on multi-party TV-show

dialogue with very different structural characteristics. However, two scope limitations remain. First, the specific saturation $K$ values are corpus-dependent: IEMOCAP saturates at $K$=10–30, MELD effectively at $K$=0, and DailyDialog or other spontaneous corpora may exhibit yet different operating points. Second, the identity of the most context-dependent emotion can shift across corpora (Sad in IEMOCAP, Anger in MELD), which we attribute to the interactional dynamics of each corpus rather than to a universal emotion-context mapping. Therefore, the observed context-length effects and emotion-specific dependencies should be understood as corpus-sensitive patterns, not as universal properties of emotional discourse.

**Operationalization choices in context and discourse analysis.** Our context analysis uses turn-based windows and a fixed definition of "past-only" context, and our discourse analysis relies on normalized token positions with predefined periphery thresholds (LP < 0.15, RP > 0.85). Alternative operationalizations (e.g., sentence- or time-based windows, speaker-conditioned windows, syntactic peripheries, or discourse-unit segmentation) may yield different quantitative estimates. Future work should stress-test these choices and assess whether conclusions hold under reasonable variants.

**Closed discourse-marker inventory.** We analyze a predefined inventory of 20 markers drawn from established taxonomies. While theory-driven, this closed set may miss informal and multiword pragmatic expressions prevalent in conversational speech. A promising direction is to complement this with data-driven discovery (e.g., collocation-based mining, pragmatic phrase induction, or supervised taggers) and to examine whether automatically discovered marker families show stronger or more generalizable emotion associations.

**Broader impact and ethical considerations.** Several aspects of this work motivate explicit discussion of potential broader impact. First, our applications-oriented framing references sensitive domains such as mental-health support and empathetic conversational agents. The findings in this paper, however, characterise probabilistic tendencies in emotional discourse rather than reliable categorical judgements. Some categories are also weakly identified from text alone (e.g., Happy versus Excited; Section 4.4). In high-stakes deployments, surfacing such categorical labels directly to end-users without appropriate calibration and human oversight risks inappropriate or even harmful system responses, and we discourage downstream use of the recognised labels as standalone clinical or affective signals. Second, our discourse-marker analysis is grounded in English-language IEMOCAP transcripts. Marker inventories, periphery norms, and emotional pragmatics vary substantially across languages, dialects, and social groups; applying our positional generalisations beyond this scope without re-validation risks systematic bias and misinterpretation of emotional states. Third, while we frame contributions in terms of recognition, real-time causal ERC has potential dual-use implications: systems designed for socially aware interaction could also be repurposed for continuous and non-consensual emotional monitoring of users. Responsible deployment therefore requires transparent disclosure to users, explicit informed consent, and proportionate use, particularly in asymmetric-power settings such as workplaces, classrooms, and care contexts.

## 6  Conclusion

We presented a systematic analysis of emotion recognition in conversation (ERC) that targets two gaps in the literature: (i) clarifying which modeling choices materially affect recognition under a strictly causal (past-only) setting, and (ii) identifying interpretable discourse-level patterns that connect recognition findings to the linguistic structure of conversational discourse.

For recognition, our experiments yield three primary takeaways. First, conversational context is the dominant driver of performance: most of the attainable gain is achieved with a relatively short window of preceding turns, after which returns diminish. Second, intra-utterance hierarchical sentence encoding can be beneficial when no conversational context is available, but its advantage largely disappears once turn-level context is introduced, suggesting that inter-turn dynamics can substitute for fine-grained within-utterance structure in causal ERC. Third, integrating an external affective lexicon (SenticNet) provides no improvement in our setting, consistent with the view that modern pretrained encoders already capture much of the affective semantics needed for classification.

For linguistic analysis, our corpus study of 5,286 discourse-marker occurrences reveals small but reliable emotion-specific positional tendencies. In particular, Sad utterances exhibit reduced left-periphery marker usage relative to other emotions, which we interpret as diminished overt turn-management signaling. This observation complements the recognition-side finding that Sad benefits most from added context: when explicit pragmatic cues are weaker, conversational history becomes more important for disambiguation.

Finally, our confusion analysis highlights that, in text-only ERC, some fine-grained distinctions in the 6-way taxonomy are weakly identified without prosodic information (e.g., asymmetric Happy–Excited confusions). Taken together, these results suggest practical guidance for causal ERC system design: simple flat encoders with moderate past context can capture most predictive information; more elaborate utterance structure offers limited marginal benefit once context is available; and careful attention to discourse phenomena offers a principled path toward interpretable analyses of how emotions are expressed in conversation. Whether the linguistic patterns we report can be operationalised as cues for emotion-conditioned dialogue generation is an open question that we leave to future work; our contribution here is empirical and descriptive rather than generative.

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

## A  Discourse Marker Inventory

Our discourse marker analysis uses markers drawn from established taxonomies in discourse and pragmatics research. We compiled a search inventory from Schiffrin (1987), Fraser (1999), Traugott (2010), Verhagen (2005), Aijmer (2013), Beeching & Detges (2014), and Biber & Finegan (1989). Table 7 lists the 20 markers that were empirically found in IEMOCAP, along with their frequencies and theoretical sources.

Table 7: Discourse markers found in IEMOCAP with occurrence counts and source references.

| Marker | Category | Count | Source |
|---|---|---|---|
| and | Elaborative | 2,372 | Schiffrin (1987); Fraser (1999) |
| so | Inferential | 1,968 | Schiffrin (1987); Fraser (1999); Beeching & Detges (2014) |
| like | Pragmatic particle | 1,210 | Aijmer (2013) |
| but | Contrastive | 760 | Schiffrin (1987); Fraser (1999); Beeching & Detges (2014) |
| well | Turn-management | 727 | Schiffrin (1987); Beeching & Detges (2014) |
| oh | Turn-management | 564 | Schiffrin (1987); Beeching & Detges (2014) |
| you know | Intersubjective | 393 | Schiffrin (1987); Verhagen (2005) |
| i mean | Intersubjective | 240 | Schiffrin (1987); Verhagen (2005) |
| maybe | Epistemic (doubt) | 195 | Traugott (2010); Biber & Finegan (1989) |
| though | Contrastive | 162 | Fraser (1999); Beeching & Detges (2014) |
| i think | Epistemic (stance) | 131 | Traugott (2010) |
| probably | Epistemic (doubt) | 83 | Traugott (2010); Biber & Finegan (1989) |
| i guess | Epistemic (stance) | 77 | Traugott (2010) |
| yet | Contrastive | 29 | Fraser (1999) |
| also | Elaborative | 18 | Fraser (1999) |
| i believe | Epistemic (stance) | 10 | Traugott (2010) |
| however | Contrastive | 6 | Fraser (1999) |
| although | Contrastive | 5 | Fraser (1999) |
| unfortunately | Attitudinal | 4 | Biber & Finegan (1989) |
| therefore | Inferential | 1 | Fraser (1999) |
| **Total** | | **8,955** | |

## B  Hyperparameter Sensitivity Analysis

To ensure fair comparison, we verified that alternative hyperparameter choices do not yield statistically significant performance differences. We conducted 14 pairwise comparisons across layer selection and pooling methods using paired t-tests and Friedman tests with 10 random seeds.

### B.1  Layer Selection: `last` vs `avg_last4`

Table 8 presents weighted F1 scores (%) for different layer extraction methods at utterance-level (K=0). None of the six comparisons showed significant differences (all $p > 0.24$).

Table 8: Layer comparison at utterance-level (paired t-test, $n = 10$)

| Task | Pooling | last | avg_last4 | $p$-value |
|---|---|---|---|---|
| 4-way | mean | $64.89 \pm 0.79$ | $64.63 \pm 0.91$ | 0.382 |
| 4-way | wmean_pos | $64.61 \pm 1.06$ | $64.30 \pm 0.94$ | 0.330 |
| 4-way | wmean_pos_rev | $64.65 \pm 1.23$ | $64.54 \pm 0.73$ | 0.753 |
| 6-way | mean | $52.30 \pm 0.91$ | $52.11 \pm 0.81$ | 0.677 |
| 6-way | wmean_pos | $52.42 \pm 1.01$ | $52.05 \pm 1.04$ | 0.508 |
| 6-way | wmean_pos_rev | $52.25 \pm 0.77$ | $51.71 \pm 0.89$ | 0.244 |

## B.2 Pooling Method Comparison

Table 9 shows performance across three pooling methods using the Friedman test. No significant differences were found at either utterance-level or turn-level (all $p > 0.08$).

Table 9: Pooling method comparison (Friedman test, $n = 10$)

| Level | Task | mean | wmean_pos | wmean_pos_rev | $p$-value |
|---|---|---|---|---|---|
| *Utterance-level (K=0), layer=last* | | | | | |
| | 4-way | 64.89 | 64.61 | 64.65 | 0.670 |
| | 6-way | 52.30 | 52.42 | 52.25 | 0.905 |
| *Utterance-level (K=0), layer=avg_last4* | | | | | |
| | 4-way | 64.63 | 64.30 | 64.54 | 0.123 |
| | 6-way | 52.11 | 52.05 | 51.71 | 0.082 |
| *Turn-level (best K per seed)* | | | | | |
| | 4-way | $82.69 \pm 0.50$ | $82.49 \pm 0.46$ | $82.37 \pm 0.67$ | 0.301 |
| | 6-way | $66.88 \pm 0.84$ | $67.07 \pm 0.69$ | $66.57 \pm 0.48$ | 0.150 |

## B.3 Hierarchical Aggregation Comparison

For hierarchical encoding at turn-level, we compared aggregation methods (Table 10). No significant differences were observed (all $p > 0.17$).

Table 10: Hierarchical aggregation comparison (paired t-test, $n = 10$)

| Task | mean | wmean_pos | $t$-statistic | $p$-value |
|---|---|---|---|---|
| 4-way | $81.89 \pm 0.41$ | $81.57 \pm 0.56$ | 1.39 | 0.197 |
| 6-way | $66.73 \pm 0.87$ | $66.19 \pm 0.66$ | 1.47 | 0.175 |

## B.4 Summary

Table 11 summarizes all 14 comparisons. None showed statistically significant differences ($p < 0.05$), justifying our reporting of only the best-performing configurations in the main results.

Table 11: Summary of hyperparameter sensitivity analysis

| Category | # Tests | Significant | Min $p$ |
|---|---|---|---|
| Layer (last vs avg_last4) | 6 | 0/6 | 0.244 |
| Pooling (utterance-level) | 4 | 0/4 | 0.082 |
| Pooling (turn-level FLAT) | 2 | 0/2 | 0.150 |
| Aggregation (turn-level HIER) | 2 | 0/2 | 0.175 |
| **Total** | **14** | **0/14** | **0.082** |

## C    SenticNet Ablation Studies

Table 12 presents the complete results of SenticNet fusion experiments across 36 configurations.

Table 12: Complete SenticNet Fusion Results (36 Configurations). $\Delta$ shows performance change in percentage points. Each configuration evaluated with 10 seeds.

| Encoder | Task | $\alpha$ | Base | +Sentic | $\Delta(\%)$ | $p$ |
|---|---|---|---|---|---|---|
| BERT | 4-way | 0.05 | .633 | .628 | $-0.50^*$ | .029 |
| BERT | 4-way | 0.10 | .633 | .627 | $-0.57^*$ | .013 |
| BERT | 4-way | 0.20 | .633 | .627 | $-0.58^{**}$ | .008 |
| BERT | 4-way | 0.50 | .633 | .628 | $-0.49^*$ | .022 |
| BERT | 4-way | 1.00 | .633 | .628 | $-0.52^{**}$ | .007 |
| BERT | 4-way | concat | .633 | .628 | $-0.52^{**}$ | .007 |
| BERT | 6-way | 0.05 | .485 | .482 | $-0.23$ | .298 |
| BERT | 6-way | 0.10 | .485 | .482 | $-0.26$ | .282 |
| BERT | 6-way | 0.20 | .485 | .481 | $-0.35$ | .146 |
| BERT | 6-way | 0.50 | .485 | .483 | $-0.14$ | .537 |
| BERT | 6-way | 1.00 | .485 | .484 | $-0.10$ | .623 |
| BERT | 6-way | concat | .485 | .484 | $-0.10$ | .623 |
| RoBERTa | 4-way | 0.05 | .634 | .633 | $-0.09$ | .705 |
| RoBERTa | 4-way | 0.10 | .634 | .634 | $+0.00$ | .999 |
| RoBERTa | 4-way | 0.20 | .634 | .634 | $+0.01$ | .958 |
| RoBERTa | 4-way | 0.50 | .634 | .634 | $-0.00$ | .996 |
| RoBERTa | 4-way | 1.00 | .634 | .633 | $-0.07$ | .804 |
| RoBERTa | 4-way | concat | .634 | .633 | $-0.07$ | .804 |
| RoBERTa | 6-way | 0.05 | .487 | .487 | $+0.05$ | .851 |
| RoBERTa | 6-way | 0.10 | .487 | .488 | $+0.07$ | .794 |
| RoBERTa | 6-way | 0.20 | .487 | .486 | $-0.04$ | .895 |
| RoBERTa | 6-way | 0.50 | .487 | .487 | $+0.01$ | .981 |
| RoBERTa | 6-way | 1.00 | .487 | .487 | $+0.06$ | .777 |
| RoBERTa | 6-way | concat | .487 | .487 | $+0.06$ | .777 |
| S-RoBERTa | 4-way | 0.05 | .656 | .653 | $-0.28$ | .230 |
| S-RoBERTa | 4-way | 0.10 | .656 | .653 | $-0.32$ | .231 |
| S-RoBERTa | 4-way | 0.20 | .656 | .653 | $-0.34$ | .205 |
| S-RoBERTa | 4-way | 0.50 | .656 | .653 | $-0.29$ | .268 |
| S-RoBERTa | 4-way | 1.00 | .656 | .654 | $-0.25$ | .385 |
| S-RoBERTa | 4-way | concat | .656 | .654 | $-0.25$ | .385 |
| S-RoBERTa | 6-way | 0.05 | .516 | .516 | $+0.08$ | .794 |
| S-RoBERTa | 6-way | 0.10 | .516 | .516 | $+0.07$ | .815 |
| S-RoBERTa | 6-way | 0.20 | .516 | .516 | $+0.04$ | .884 |
| S-RoBERTa | 6-way | 0.50 | .516 | .515 | $-0.01$ | .963 |
| S-RoBERTa | 6-way | 1.00 | .516 | .516 | $+0.02$ | .925 |
| S-RoBERTa | 6-way | concat | .516 | .516 | $+0.02$ | .925 |

$^*p < .05$, $^{**}p < .01$ (paired $t$-test). Fusion: $\mathbf{e} = (1 - \alpha)\mathbf{e}_{\text{ctx}} + \alpha\mathbf{e}_{\text{sentic}}$.

