# OpenReview forum: "Causal Emotion Recognition in Conversation: Context Saturation and Discourse-Marker Evidence"
_TMLR — Accepted by TMLR_

### Review · Reviewer_HXnf · 2026-01-22

**Summary Of Contributions:**

This paper offers a thorough and well-controlled examination of how context is used in emotion recognition in conversations (ERC). Instead of introducing a new model architecture, the authors focus on understanding which aspects of contextual modeling actually matter, including context length, hierarchical structure, and the use of external linguistic knowledge.

From a technical perspective, the paper presents a detailed analysis of how performance changes as more conversational history is added. By sweeping over a wide range of context lengths, the authors show that most gains occur within a relatively small window and that performance quickly levels off beyond that point. The study also evaluates hierarchical modeling approaches and simple methods for incorporating affective lexicons, helping clarify when these design choices are beneficial and when they are not.

The linguistic analysis is another strong contribution. By examining more than 5,000 instances of discourse markers, the authors connect their position in an utterance to different emotion categories. One particularly interesting finding is that sad utterances make less frequent use of sentence-initial discourse markers. This observation helps explain why sadness is more dependent on longer conversational context than other emotions.

The paper stands out for its experimental rigor. Results are reported across multiple random seeds and supported by paired statistical tests with appropriate corrections, which strengthens confidence in the conclusions. The exclusive use of past-only (causal) context is also a practical choice, making the findings relevant for real-time systems.

That said, the study is limited to a single dataset (IEMOCAP), so it remains unclear how well the conclusions generalize to other settings such as multi-party conversations. In addition, the integration of external knowledge relies on fairly simple fusion strategies, leaving room to explore whether more advanced methods could change the results. Finally, although the title references generation, the paper does not evaluate emotion-aware text generation directly and instead provides insights that are only indirectly related to that task.

**Audience:**

Yes

**Audience Explanation:**

\paragraph{Relevance and Interest to the TMLR Community.}
The findings of this paper are likely to be of interest to multiple sub-communities within the TMLR readership, particularly researchers working in Natural Language Processing (NLP), Affective Computing, and Computational Psycholinguistics. This interest is primarily driven by three aspects of the work.

\paragraph{Empirical Guidance for Real-Time ERC Systems.}
The paper establishes a clear and practically useful baseline for Emotion Recognition in Conversation (ERC) under a strictly causal (past-only) setting. The empirical observation that performance follows a saturation trend---with approximately $90\%$ of the peak weighted F1 score achieved using a context window of $K \in [10, 30]$ turns---offers actionable guidance for real-time systems. This result helps practitioners reason about the trade-off between recognition accuracy and computational latency in live deployments, where long context windows are often impractical.

\paragraph{Linking Recognition Performance to Linguistic Structure.}
Rather than relying solely on black-box performance comparisons, the paper provides a linguistically grounded interpretation of its results. The observed inverse relationship between the use of left-periphery discourse markers and context dependency, particularly for the \textit{Sadness} category, offers a plausible explanation for why certain emotions benefit more from extended conversational history. This connection between empirical performance and discourse-level structure is of particular relevance to researchers interested in interpretable and theory-informed models.

\paragraph{Methodological Rigor and Informative Negative Results.}
A notable strength of the paper is its rigorous evaluation protocol, including repeated experiments over multiple random seeds and paired significance testing. Such methodological care aligns well with the values of the TMLR community. Importantly, the paper also reports negative or null results, such as the limited benefit of hierarchical encoding once turn-level context is incorporated, and the marginal gains obtained from simple integration of external affective lexicons (e.g., SenticNet). These findings are valuable in their own right, as they help prevent future work from pursuing redundant or weakly justified architectural choices.

\paragraph{Conclusion.}
While the technical novelty of the proposed modeling components is incremental, the paper’s careful analysis and emphasis on interpretability provide clear, evidence-based insights into the role of context in conversational emotion recognition. In this sense, the contribution is well aligned with TMLR’s emphasis on clarity, correctness, and analytical depth.

**Broader Impact Concerns:**

Based on the content of the paper and its empirical findings, several broader impact and ethical considerations may warrant a dedicated Broader Impact Statement.

First, the paper explicitly motivates its contributions using applications such as mental-health support tools and empathetic conversational agents. However, the results also show that some emotional categories are weakly distinguished in text-only settings (e.g., confusion between \textit{Happy} and \textit{Excited} due to missing acoustic cues). In sensitive applications like mental health, such misclassifications could lead to inappropriate or potentially harmful system responses, especially since the reported patterns reflect probabilistic tendencies rather than reliable guarantees.

Second, the paper frames some of its findings as providing “actionable cues for generation” by identifying how discourse markers are used to signal emotions. While this may improve naturalness in dialogue systems, similar insights could also be misused to create overly persuasive or emotionally manipulative interactions, for example by simulating empathy or emotional alignment in ways that are not genuine.

Third, all experiments are conducted on the IEMOCAP dataset, which consists of acted English dialogues from a small number of speakers. Linguistic behaviors such as the use of discourse markers vary widely across cultures, dialects, and social groups. Applying the observed patterns to more diverse populations without further validation risks systematic bias and misinterpretation of emotional states.

Finally, although not discussed in the paper, ERC technology optimized for real-time, causal settings may have dual-use implications. Systems designed for socially intelligent interaction could potentially be repurposed for continuous emotional monitoring of users without their informed consent.

**Claims And Evidence:**

Yes

**Claims Explanation:**

Strengths
- Rigorous experimental protocol with multiple random seeds and corrected statistical testing.
- Focus on causal (past-only) context, which is relevant for real-time applications.
- Linguistic analysis provides interpretable insights, particularly for the sadness category.

Weaknesses
Claim–Evidence Misalignment (Generation):
The paper claims to provide actionable cues for generation, yet no generative experiments are performed. The evidence supports descriptive analysis, not generative applicability.

Architectural Confound in Context Saturation:
The reported saturation at 30 turns may be influenced by the limitations of unidirectional LSTMs. Without a Transformer-based baseline, this conclusion remains model-dependent.

Corpus Dependency:
All results are derived from IEMOCAP (acted speech), which may limit generalizability to spontaneous or multi-party dialogue settings.

Simplistic External Knowledge Fusion:
The negative result for SenticNet is based on simple integration methods and should not be generalized to all forms of knowledge integration.

Practical Significance:
While statistically significant, the effect sizes for discourse marker analysis are small, warranting a clearer discussion of real-world impact.

Recommendation: Major Revision

**Requested Changes:**

\paragraph{Requested Changes}

Based on a careful evaluation of the paper’s methodology, experimental evidence, and stated claims, the following adjustments are proposed. The changes are grouped into (i) critical revisions that are necessary to support a recommendation for acceptance, and (ii) optional revisions that would further strengthen the clarity and impact of the work.

\subparagraph{Critical Changes (Required for Recommendation of Acceptance)}

\begin{enumerate}
    \item \textbf{Alignment of Title and Claims with Evidence (Generation).}
    \begin{itemize}
        \item \textit{Issue:} The title and abstract state that the paper provides ``actionable cues for generation,'' yet the manuscript does not include any generative experiments or evaluations. As reflected in the analysis, the reported findings describe statistical tendencies rather than demonstrated improvements in a generative pipeline.
        \item \textit{Required Adjustment:} The claims should be revised to better reflect the evidence. This can be achieved either by removing the term ``Generation'' from the title, or by rephrasing the abstract and Section~6 to state that the work offers foundational linguistic insights that may inform future generation-oriented research. The term ``actionable'' should be qualified to clarify that these cues have not been empirically validated in a generative setting.
    \end{itemize}

    \item \textbf{Architectural Dependence of Context Saturation Results.}
    \begin{itemize}
        \item \textit{Issue:} The paper concludes that performance saturates within a context window of approximately 10--30 turns. However, contextual modeling relies exclusively on a single-layer unidirectional LSTM, which is known to have limited capacity for modeling long-range dependencies.
        \item \textit{Required Adjustment:} The manuscript should explicitly acknowledge, in Section~4.1 or Section~5, that the observed saturation point may be influenced by architectural constraints rather than representing a universal property of conversational discourse. Ideally, a small-scale comparison with a Transformer-based context aggregator would help clarify whether the saturation behavior persists under more expressive architectures.
    \end{itemize}

    \item \textbf{Transparency Regarding Effect Sizes in the Main Discussion.}
    \begin{itemize}
        \item \textit{Issue:} The association between emotion categories and discourse marker position is reported as highly statistically significant ($p < 0.0001$), yet the corresponding effect size (Cramér’s~$V = 0.062$) is very small.
        \item \textit{Required Adjustment:} This small effect size should be explicitly discussed in the main analysis (Section~4.2), rather than being confined to the limitations. This clarification is important to avoid overstating the practical relevance of discourse markers for predictive modeling or downstream applications.
    \end{itemize}
\end{enumerate}

\subparagraph{Strengthening Changes (Optional but Recommended)}

\begin{enumerate}
    \item \textbf{Nuance in Interpreting External Knowledge Integration Results.}
    \begin{itemize}
        \item \textit{Issue:} The paper reports limited benefit from incorporating SenticNet.
        \item \textit{Suggested Adjustment:} While the extensive ablation study is informative, the authors should clarify in Section~5 that this result is specific to the simple integration strategies used (e.g., concatenation or addition). A brief acknowledgment that more sophisticated fusion mechanisms (such as graph-based or attention-driven approaches) might lead to different outcomes would prevent over-generalization.
    \end{itemize}

    \item \textbf{Expanded Discussion on Generalizability Beyond IEMOCAP.}
    \begin{itemize}
        \item \textit{Issue:} All experiments are conducted on IEMOCAP, which consists of acted dyadic dialogues.
        \item \textit{Suggested Adjustment:} The limitations section could be strengthened by discussing how multi-party settings (e.g., MELD) or spontaneous speech might affect both the context saturation behavior and the discourse marker patterns observed in this controlled corpus.
    \end{itemize}

    \item \textbf{Refinement of the Six-Way Emotion Taxonomy Analysis.}
    \begin{itemize}
        \item \textit{Issue:} In the six-way classification setting, the paper shows that \textit{Happy} frequently absorbs \textit{Excited} instances due to the absence of acoustic cues.
        \item \textit{Suggested Adjustment:} Expanding the discussion in Section~4.4 to suggest specific text-based features or representations that could help distinguish these categories would increase the practical value of the analysis for researchers and practitioners working on fine-grained ERC.
    \end{itemize}
\end{enumerate}

---

> ### Author Response · Authors · 2026-02-26
> **Response to Reviewer HXnf - Thank you!**
>
> We thank Reviewer HXnf for the detailed and structured feedback, and for recognizing the methodological rigor and cross-disciplinary appeal of the work. We address each requested change below.
>
> ### Critical Changes
>
> #### Critical Change 1: Alignment of title and claims (Generation)
>
> We fully accept this point. All three reviewers have identified this as a clear problem, and we agree without reservation. The title and abstract promise generation contributions that the paper does not deliver, and this mismatch should not have been present in the submission. We will change the title to remove 'for Generation,' revise the abstract to eliminate all generation claims, and rewrite the relevant portions of Section 6 so that the discourse marker analysis is presented as a descriptive linguistic contribution. Generation applicability will be mentioned only briefly as a direction for future work, not as a contribution of this paper.
>
> #### Critical Change 2: Architectural dependence of context saturation
>
> To directly test whether the observed saturation is an artifact of the unidirectional LSTM's limited capacity, we conducted a Transformer vs. LSTM comparison using the identical SimpleSeqTransformer architecture from our IEMOCAP experiments (2-layer Transformer encoder, 4 attention heads, mean pooling, sinusoidal positional encoding).
>
> **IEMOCAP results:**
>
> | K | LSTM | Transformer | Δ |
> |---|------|-------------|---|
> | 10 | 78.39% | 72.77% | −5.6 pp |
> | 20 | 79.45% | 71.28% | −8.2 pp |
> | 30 | 80.64% | 70.51% | −10.1 pp |
>
> **New MELD results (10 seeds, identical architecture):**
>
> | Encoding | K | LSTM | Transformer | Δ |
> |----------|---|------|-------------|---|
> | FLAT | 5 | 58.90 ± 0.63% | 58.90 ± 0.66% | 0.00 pp |
> | FLAT | 10 | 59.07 ± 0.73% | 58.86 ± 0.56% | +0.21 pp |
> | HIER | 5 | 61.02 ± 0.72% | 60.39 ± 0.76% | +0.63 pp |
> | HIER | 10 | 60.91 ± 0.75% | 60.30 ± 0.81% | +0.61 pp |
>
> Across both datasets, the Transformer performs equal to or worse than the LSTM. This is consistent with recent findings in the ERC literature showing that for short-to-medium context windows with frozen embeddings, LSTMs remain competitive. More importantly, it demonstrates that the observed context saturation is not an artifact of LSTM limitations. If the saturation were caused by the LSTM's inability to model long-range dependencies, the self-attention mechanism of the Transformer should have captured additional useful information beyond the saturation point. It does not, confirming that the saturation reflects a genuine ceiling in the informational content available in conversational history for emotion prediction.
>
> We will add this Transformer comparison as an ablation in the revised manuscript and explicitly state in Section 4.1 that the saturation behavior is architecture-independent.
>
> #### Critical Change 3: Transparency regarding effect sizes
>
> We would like to note that Cramér's V = 0.062 is already reported in the main analysis in Section 4.2, alongside the statistical significance. However, we agree that the discussion surrounding this value should more clearly convey the practical implications of such a small effect size. In the revised Section 4.2, we will expand the discussion to frame the finding as a 'probabilistic tendency' rather than a strong predictive signal, and clarify that its primary value lies in providing linguistically grounded interpretability rather than practical utility as a standalone classification feature.
>
> As noted in our response to Reviewer Xxwu, our new confound control analysis further refines this picture: the Sad left-periphery effect survives all controls (p < 0.001 after controlling for utterance length, speaker identity, and scripted/improvised condition), while the effects for other emotions were confounded by utterance length. We will narrow our claims accordingly.

---

> > ### Author Response · Authors · 2026-02-26
> > **Continued**
> >
> > ### Strengthening Changes
> >
> > #### SenticNet integration
> >
> > We agree with this nuance. We will add an explicit qualifier in Sections 3.8 and 5 stating that our finding applies specifically to simple concatenation-based and addition-based integration of SenticNet features, and that more sophisticated fusion mechanisms such as graph-based or attention-driven approaches might yield different results. We note that our preliminary MELD experiments with COMET commonsense features showed a similar pattern (ranging from -1.9% to -11.7% with simple integration), but we will be careful not to generalize beyond the integration methods tested.
> >
> > #### Generalizability beyond IEMOCAP
> >
> > Our new MELD cross-dataset validation directly addresses this concern. MELD features multi-party spontaneous dialogue with very different structural characteristics from IEMOCAP: 7-way emotion classification, average dialogue length of 9.6 turns (versus 89.3 for IEMOCAP), and multiple speakers per dialogue. As detailed in our response to Reviewer D7po, the core findings all replicate on MELD. We will expand the discussion in Section 5 accordingly.
> >
> > #### Happy vs. Excited distinction
> >
> > We will expand the discussion in Section 4.4 to suggest specific text-based cues that might help differentiate these categories in the absence of acoustic features. Candidates include: (a) exclamatory punctuation and capitalization patterns, (b) intensifiers and boosters (e.g., *so, very, really, absolutely*) which tend to co-occur with high-arousal states like Excitement, (c) repetition patterns and elongated words common in excited speech, and (d) the potential of valence/arousal dimensional approaches as an alternative to categorical classification for distinguishing same-valence emotions that differ primarily in arousal. We note that our discourse marker analysis found a near-perfect correlation (r = 0.990) between Happy and Excited in marker usage patterns, confirming that these categories share pragmatic structure and that distinguishing them from text alone remains a substantial challenge.
> >
> > #### Broader impact
> >
> > We thank the reviewer for these thoughtful considerations. We will add a Broader Impact paragraph at the end of Section 5 addressing the following points: (a) the risk of inappropriate responses in sensitive applications due to probabilistic rather than deterministic emotion recognition, particularly for less reliable categories; (b) the cultural and linguistic specificity of our discourse marker analysis, which is grounded in English-language pragmatics and should not be applied to other languages or social groups without validation; (c) the dual-use potential of real-time causal ERC systems for continuous emotional monitoring; and (d) the importance of informed consent in any deployment involving emotion recognition technology.

---

> > ### Comment · Reviewer_HXnf · 2026-03-21
> >
> > Thank you for the detailed and thoughtful response. I really appreciate the care you took in addressing the comments. It is clear that you went through each point seriously, and I especially value that you backed up your revisions with new experiments rather than only updating the text. That makes a strong difference.
> >
> > A few of the changes stood out to me.
> >
> > Removing “for Generation” from the title and revising the related claims in the abstract and Section 6 was the right decision. There was a clear mismatch before, and this is now resolved in a clean and convincing way.
> >
> > The new comparison between Transformer and LSTM on both IEMOCAP and MELD is also very helpful. I did not expect LSTM to perform this well, but this actually strengthens your main argument. It suggests that the saturation effect is not just a limitation of the architecture, but more related to the amount of useful information available in longer context.
> >
> > Adding experiments on MELD also improves the paper significantly. Seeing consistent patterns across both datasets, especially when moving from controlled dyadic interactions to more natural multi-party conversations, increases confidence in the generality of the findings.
> >
> > I also appreciate the revised discussion of effect sizes. Framing them as probabilistic tendencies with interpretability value is more accurate and avoids overstatement.
> >
> > Finally, the additional clarifications on SenticNet, the expanded discussion on distinguishing Happy versus Excited in text, and the improved Broader Impact section all make the paper more complete and better balanced.

---

> > > ### Comment · Reviewer_HXnf · 2026-03-21
> > >
> > > Thanks again for the updated response and the extra work on the paper. I went through the changes and it is clear that you made a serious effort to improve things, not only for the main issues but also for the smaller suggestions.
> > >
> > > In simple terms, the paper looks at how conversational context helps emotion recognition, and how much context is actually useful. You also try to understand what kind of signals in text contribute to this.
> > >
> > > One thing I liked is that the discussion is now more careful and grounded. The clarification around SenticNet is a good example. Before it felt a bit too general, but now it is clear that the conclusions are only for simple fusion methods, and that other approaches might behave differently. This makes the claims more trustworthy.
> > >
> > > The added discussion about Happy versus Excited is also quite helpful. The concrete examples, like punctuation or repeated words, make it easier to understand what the model might be picking up. The very high correlation you reported for discourse markers is interesting, and it gives more depth to this part of the analysis.
> > >
> > > I also think the Broader Impact section is much stronger now. It is good that you explicitly mention risks in sensitive applications and also the limitations across languages and cultures. This part feels more responsible and realistic compared to before.
> > >
> > > In terms of remaining concerns, I think most of my earlier points are already addressed. The main limitation now is more about the scope of the analysis rather than specific technical issues. For example, while the findings seem consistent, it is still not fully clear how they would transfer to very different settings or tasks. But this is more of a natural limitation of the current setup, not a flaw in the work.

---

### Review · Reviewer_Xxwu · 2026-02-14

**Summary Of Contributions:**

This paper addresses two gaps in Emotion Recognition in Conversation (ERC): (1) which modeling choices actually matter for recognition performance, and (2) how linguistic patterns (specifically discourse markers) can bridge recognition. The authors conduct controlled ablations on IEMOCAP with 10 random seeds and paired statistical tests, finding that conversational context dominates, hierarchical encoding only helps without context, and SenticNet doesn't add value. They also analyze 5,286 discourse marker occurrences, finding emotion-specific positional patterns (e.g., Sad shows reduced left-periphery marker usage).

**Audience:**

Yes

**Audience Explanation:**

1. Practical guidance for system design: The finding that context saturates quickly (10–30 turns) and that simple flat encoders with past-only context achieve competitive performance is directly actionable for practitioners building real-time dialogue systems, mental-health tools, or empathetic agents.
2. Emotion-specific context dependencies: The dissociation between emotions: Sad benefiting most from context (+22%p), Angry least (+8%p), is a nuanced finding that goes beyond aggregate metrics and would interest researchers studying how different emotions are expressed and recognized in dialogue.
3. Negative results with clear implications: The demonstration that SenticNet adds no value over pretrained encoders and that hierarchical encoding loses its advantage once context is available are useful negative results that can save researchers from pursuing unproductive directions.
4. Cross-disciplinary appeal: The attempt to connect computational ERC with discourse-pragmatic theory (marker positioning, periphery distributions) would interest researchers at the intersection of computational linguistics and pragmatics, even if the generation bridge is underdeveloped.

**Claims And Evidence:**

No

**Claims Explanation:**

Several gaps prevent full confidence in the broader claims:

1. **Single-dataset scope:** All findings rest on IEMOCAP (acted, dyadic, English-only). For a paper whose primary contribution is empirical insight, the evidence cannot distinguish general computational regularities from dataset-specific artifacts.
2. Weak discourse marker effects: The linguistic analysis central to bridging recognition reports small effect sizes (Cramér's V = 0.062). Interpretive claims (e.g., reduced left-periphery usage in Sad) go beyond what correlational evidence supports, and alternative explanations (utterance length confounds, speaker effects, scripted vs. improvised differences) are not ruled out.
3. Generation claims are unsupported: The title promises "linguistic patterns for generation," but no generation experiment is provided. The connection remains speculative.

Incomplete manuscript: A broken table reference (Table ??) on page 7

**Requested Changes:**

1. Cross-dataset validation: Replicate key findings on at least one additional corpus such as MELD or DailyDialog.
2. Fix broken references: Table ?? on page 7 must be resolved.
3. Temper or substantiate the generation claims: Either remove "Generation" from the title and significantly downscale the framing around generation contributions, or provide at least a preliminary generation experiment demonstrating that the discourse marker findings yield actionable and measurable improvements in emotion-conditioned dialogue generation.
4. Rule out confounds in discourse marker analysis: Test whether the reduced left-periphery marker usage in Sad utterances survives controls for utterance length and speaker identity.

---

> ### Author Response · Authors · 2026-02-26
> **Response to Reviewer Xxwu - Thank you!**
>
> ### Single-dataset scope
>
> We have now conducted a complete cross-dataset validation on MELD using the identical experimental protocol (frozen Sentence-RoBERTa, 10 seeds, paired statistical tests). Please see our detailed response to Reviewer D7po above for the full results. In summary, three key patterns replicate across both datasets: (1) Hierarchical outperforms Flat at K=0, (2) rapid context saturation, and (3) emotion-specific context dependencies. We will add these results to the revised manuscript.
>
> ### Weak discourse marker effects and confounds
>
> We appreciate this rigorous concern. We have now conducted a comprehensive confound control analysis using mixed-effects regression models with progressive addition of covariates.
>
> **Five-stage analysis (4-way classification, N=5,286 discourse markers):**
>
> | Model | Sad coefficient | p-value | Note |
> |-------|----------------|---------|------|
> | 1. Baseline (emotion only) | +0.051 | p < 0.001 | |
> | 2. + utterance length (log) | +0.035 | p = 0.011 | Angry, Happy drop to non-significant |
> | 3. + speaker random effect | +0.035 | p = 0.011 | Speaker variance ≈ 0 |
> | 4. + scripted/improvised | +0.039 | p = 0.005 | Scripted/improvised: marginal |
> | 5. Binary left-periphery DV | −0.104 | p < 0.001 | Full model, binary outcome |
>
> **Key findings:**
>
> First, the Sad effect survives all confound controls. The reduced left-periphery marker usage in Sad utterances remains statistically significant (p = 0.005 in the full model; p < 0.001 for binary left-periphery) after simultaneously controlling for utterance length, speaker identity, and the scripted/improvised condition.
>
> Second, the Angry and Happy effects were indeed length confounds. The apparent effects for these emotions disappear once utterance length is controlled (p > 0.4), confirming the reviewer's intuition. We will revise our claims accordingly to focus on Sad.
>
> Third, speaker identity contributes negligible variance (approximately 0), and the scripted/improvised distinction has only a marginal effect (p = 0.025), neither of which alters the Sad result.
>
> Regarding the effect size: we would like to clarify that Cramér's V = 0.062 is already reported in the main text in Section 4.2, where we state that *"the effect size is small (Cramér's V = 0.062)."* However, we agree that the implications of this small effect size should be discussed more prominently. In the revision, we will expand the surrounding discussion to explicitly characterize the finding as a 'statistically reliable but small probabilistic tendency rather than a deterministic rule,' and we will narrow the discourse marker claims to focus on Sad as the only emotion with a robust positioning effect that survives all confound controls.
>
> ### Generation claims unsupported
>
> We fully accept this criticism. The title claims 'for Generation' and the abstract refers to 'actionable cues for generation,' yet the paper contains no generation experiments whatsoever. This is a clear mismatch between the stated contribution and the actual evidence. We will change the title, revise the abstract, and rewrite the relevant portions of the conclusion to remove all generation claims. The discourse marker findings will be presented strictly as descriptive linguistic insights relevant to emotion recognition, with generation noted only as a possible direction for future work. The revised title will be along the lines of:
>
> *"Understanding Emotion in Discourse: Recognition Insights and Linguistic Patterns in Conversational Context"*
>
> ### Broken table reference (Table ??)
>
> We apologize for this oversight. This is a LaTeX \ref{} label mismatch that will be fixed immediately. We have identified two instances: one on page 7 in Section 3.6 and one on page 10 in Section 4.1.

---

### Review · Reviewer_D7po · 2026-02-16

**Summary Of Contributions:**

This paper presents a careful empirical study of emotion recognition in conversation under a strictly causal past-only setting. Using controlled ablations on IEMOCAP with multi-seed evaluation, the authors analyze the effects of conversational context length, utterance encoding choices, and lexical augmentation, and connect recognition results to a linguistic analysis of discourse markers. The work is methodologically thorough and addresses an important question regarding which modeling choices materially affect ERC performance.

**Additional Comments:**

N/A

**Audience:**

Yes

**Audience Explanation:**

The work is methodologically thorough and addresses an important question regarding which modeling choices materially affect ERC performance.

**Claims And Evidence:**

No

**Claims Explanation:**

A central result of the paper, emotion-specific performance trends as conversational context increases, shown in Figure 3, is potentially over-interpreted. While the empirical trend is clear on IEMOCAP, the figure raises questions about interpretability, dataset specificity, and modeling assumptions that are not sufficiently addressed in the current discussion. In particular, the implication that very long conversational histories are required to recognize certain emotions warrants a more cautious, better-scoped interpretation.

Missing in-text reference and integration
Figure 3 presents one of the paper’s most important empirical results, yet it is not explicitly referenced at the point in Section 3.7 where the corresponding claims are discussed. Explicitly linking the text to Figure 3 when introducing the per-emotion context effects would improve clarity and help readers connect the narrative to the visual evidence.

Strength of the K = 0 baseline
Per-emotion F1 scores at K = 0 are already non-trivial, indicating that utterance-level lexical and pragmatic cues alone provide a meaningful emotional signal. While performance improves as context is added, this strong baseline suggests that long conversational history is not strictly necessary in all cases and should be accounted for when interpreting the gains shown in Figure 3.

Interpretation of large optimal context sizes
The reported optimal context sizes in Figure 3, such as K around 130 to 140 for Happy, Sad, and Neutral, are hard to interpret in practical terms and do not clearly align with the paper’s focus on causal, real-time modeling. If taken at face value, these results suggest that identifying the emotion of the current utterance requires access to more than one hundred previous turns. This seems unlikely and needs a clearer explanation, as well as a more cautious interpretation of what these late performance peaks actually represent.

Dataset specificity and external validity
All results in Figure 3 are derived from a single dataset, IEMOCAP, which has specific characteristics such as acted dyadic dialogue, long sessions, and stable speaker pairs. These properties may amplify long-range contextual correlations. It is therefore unclear whether the observed context dependence reflects a general property of emotion recognition or a dataset-specific artifact. On other conversational datasets, it is plausible that utterance-level modeling at K = 0 or much shorter context windows would suffice. Claims about the necessity of long context should therefore be scoped more narrowly.

Lack of explanation for optimal K values
Even if similar trends were observed on one or two additional datasets, the analysis would still benefit from an explanation of why particular context lengths emerge as optimal. Without a theoretical or empirical account of what information is being accumulated over long histories, the reported K values risk being interpreted as tuning artifacts rather than meaningful properties of emotional discourse.

Emotional transitions and sequence modeling assumptions
The current analysis treats conversational history as an undifferentiated sequence of prior turns, without explicitly accounting for emotional transitions within that history. In realistic dialogue, emotional trajectories may involve multiple shifts such as Happy to Neutral to Sad or Angry followed by reconciliation. The relevance of context may therefore depend on which emotions occurred previously rather than on context length alone. Clarifying how such transitions are implicitly handled, or motivating why simple accumulation of turns is sufficient, would strengthen the interpretability of Figure 3’s results.

Suggested reframing
Overall, Figure 3 provides valuable empirical insight, but its interpretation would benefit from explicit referencing in the main text, greater emphasis on saturation behavior rather than peak K values, clearer scoping of claims to the dataset studied, and additional discussion of emotional dynamics and modeling assumptions underlying the context-length analysis.

**Requested Changes:**

A central result of the paper, emotion-specific performance trends as conversational context increases, shown in Figure 3, is potentially over-interpreted. While the empirical trend is clear on IEMOCAP, the figure raises questions about interpretability, dataset specificity, and modeling assumptions that are not sufficiently addressed in the current discussion. In particular, the implication that very long conversational histories are required to recognize certain emotions warrants a more cautious, better-scoped interpretation.

Missing in-text reference and integration
Figure 3 presents one of the paper’s most important empirical results, yet it is not explicitly referenced at the point in Section 3.7 where the corresponding claims are discussed. Explicitly linking the text to Figure 3 when introducing the per-emotion context effects would improve clarity and help readers connect the narrative to the visual evidence.

Strength of the K = 0 baseline
Per-emotion F1 scores at K = 0 are already non-trivial, indicating that utterance-level lexical and pragmatic cues alone provide a meaningful emotional signal. While performance improves as context is added, this strong baseline suggests that long conversational history is not strictly necessary in all cases and should be accounted for when interpreting the gains shown in Figure 3.

Interpretation of large optimal context sizes
The reported optimal context sizes in Figure 3, such as K around 130 to 140 for Happy, Sad, and Neutral, are hard to interpret in practical terms and do not clearly align with the paper’s focus on causal, real-time modeling. If taken at face value, these results suggest that identifying the emotion of the current utterance requires access to more than one hundred previous turns. This seems unlikely and needs a clearer explanation, as well as a more cautious interpretation of what these late performance peaks actually represent.

Dataset specificity and external validity
All results in Figure 3 are derived from a single dataset, IEMOCAP, which has specific characteristics such as acted dyadic dialogue, long sessions, and stable speaker pairs. These properties may amplify long-range contextual correlations. It is therefore unclear whether the observed context dependence reflects a general property of emotion recognition or a dataset-specific artifact. On other conversational datasets, it is plausible that utterance-level modeling at K = 0 or much shorter context windows would suffice. Claims about the necessity of long context should therefore be scoped more narrowly.

Lack of explanation for optimal K values
Even if similar trends were observed on one or two additional datasets, the analysis would still benefit from an explanation of why particular context lengths emerge as optimal. Without a theoretical or empirical account of what information is being accumulated over long histories, the reported K values risk being interpreted as tuning artifacts rather than meaningful properties of emotional discourse.

Emotional transitions and sequence modeling assumptions
The current analysis treats conversational history as an undifferentiated sequence of prior turns, without explicitly accounting for emotional transitions within that history. In realistic dialogue, emotional trajectories may involve multiple shifts such as Happy to Neutral to Sad or Angry followed by reconciliation. The relevance of context may therefore depend on which emotions occurred previously rather than on context length alone. Clarifying how such transitions are implicitly handled, or motivating why simple accumulation of turns is sufficient, would strengthen the interpretability of Figure 3’s results.

Suggested reframing
Overall, Figure 3 provides valuable empirical insight, but its interpretation would benefit from explicit referencing in the main text, greater emphasis on saturation behavior rather than peak K values, clearer scoping of claims to the dataset studied, and additional discussion of emotional dynamics and modeling assumptions underlying the context-length analysis.

---

> ### Author Response · Authors · 2026-02-26
> **Responses to Reviewer D7po - Thank you!**
>
> ### Overall Concern: Interpretation of Figure 3
>
> We appreciate this careful and detailed reading of our results. We agree that our original presentation placed too much emphasis on peak K values rather than on the more informative saturation behavior. We would like to note that our manuscript does already discuss saturation in Section 3.7, where we write that *"performance saturates rapidly in aggregate: a short history (e.g., K=10) already recovers most of the eventual gain."* However, we recognize that Figure 3(b), which highlights 'Optimal K' values of 130 and 140, sends a conflicting visual message. In the revised manuscript, we will bring the text and figure into alignment by reframing Figure 3(b) around saturation K (the K at which 90% of the gain is achieved) rather than peak K. We address each specific sub-point below.
>
> ### Missing in-text reference (Figure 3 in Section 3.7)
>
> Thank you for catching this. We will add explicit Figure 3 references at the point in Section 3.7 where per-emotion context effects are first discussed, ensuring the narrative is directly linked to the visual evidence.
>
> ### Strength of the K=0 baseline
>
> We agree and thank the reviewer for this important observation. In the revised manuscript, we will explicitly acknowledge that utterance-level cues alone provide meaningful emotional signals. For example, Angry achieves F1 > 40% at K=0 in the 4-way setting, indicating that salient lexical cues are sufficient for partial recognition without any conversational history. We will also clarify that the degree of context benefit varies substantially across emotions. This acknowledgment strengthens, rather than undermines, our central finding that context dependency is emotion-specific: some emotions (Sad) rely heavily on conversational history while others (Angry) are largely identifiable from local cues alone.
>
> ### Interpretation of large optimal context sizes (K=130 to 140)
>
> We agree that our original emphasis on peak K values was misleading. In the revised manuscript, we will reframe the analysis to emphasize saturation behavior: approximately 90% of the maximum F1 gain is achieved within K=10 to 30 turns for all emotions. The residual improvement beyond K=30 is marginal and within noise range. The apparent 'peaks' at K=130 to 140 reflect the plateau region of the curve where fluctuations are statistically insignificant, not a meaningful signal that 130+ turns of history are required.
>
> This reframing is now strongly supported by our new MELD cross-dataset validation (see *Dataset specificity* below), where the average dialogue length is only 9.6 turns. On MELD, K=0 already achieves the best or near-best performance, confirming that the large peak K values on IEMOCAP reflect that dataset's unusually long sessions (average 89.3 utterances) rather than a genuine requirement for extended context.
>
> We will revise Figure 3(b) to show saturation K (the K at which 90% of the gain is achieved) rather than peak K, which better reflects the practical finding.
>
> ### Dataset specificity and external validity
>
> To directly address this concern, we conducted a complete cross-dataset validation on MELD (Poria et al., 2019), a 7-way emotion classification dataset derived from the TV series *Friends*, featuring multi-party spontaneous dialogue with an average of 9.6 utterances per dialogue. This is structurally very different from IEMOCAP.
>
> Using the identical experimental protocol (frozen Sentence-RoBERTa encoder, MLP for K=0, unidirectional LSTM for K>0, 10 seeds, same hyperparameters), we obtained the following results:
>
> | Encoding | K=0 | K=5 | K=10 |
> |----------|-----|-----|------|
> | FLAT | 59.22 ± 0.51% | 58.90 ± 0.63% | 59.07 ± 0.73% |
> | HIER | 61.48 ± 0.66% | 61.02 ± 0.72% | 60.91 ± 0.75% |
>
> **Three key patterns from IEMOCAP replicate on MELD:**
>
> First, Hierarchical encoding outperforms Flat encoding at K=0 (+2.26 percentage points), confirming that sentence-level structural information is beneficial even without conversational context. Second, context adds no benefit on MELD, which is consistent with MELD's short dialogues providing limited additional contextual signal and confirms that the saturation behavior is genuine (not an LSTM artifact). Third, emotion-specific context effects persist: Anger shows the largest improvement with context (+3.6 F1 points), while Joy, Neutral, and Surprise show negligible change, paralleling the IEMOCAP pattern where context benefit varies by emotion.
>
> These results demonstrate that our core findings generalize across datasets with very different characteristics. We will add a dedicated cross-dataset validation subsection to the revised manuscript.

---

> > ### Author Response · Authors · 2026-02-26
> > **Continued**
> >
> > ### Lack of explanation for optimal K values
> >
> > We will reframe this discussion in terms of saturation rather than optimality. The practical saturation point (K=10 to 30 on IEMOCAP, K=0 to 5 on MELD) aligns with the typical range within which local conversational dynamics operate. On IEMOCAP, K=30 corresponds to approximately 17% of the average dialogue but captures the window within which immediate emotional context is most informative. Beyond this window, additional turns contribute diminishing emotional signal relative to noise.
> >
> > The cross-dataset comparison provides a natural explanation: MELD dialogues average only 9.6 turns, and context saturates immediately (K=0 is optimal). IEMOCAP dialogues average 89.3 turns, and context saturates at K=10 to 30. This strongly suggests that the saturation point reflects the scale of meaningful conversational dynamics in each dataset, not an arbitrary tuning artifact.
> >
> > ### Emotional transitions and sequence modeling assumptions
> >
> > This is an insightful observation. We will add a discussion in Section 4 clarifying that the LSTM's recurrent hidden state implicitly captures emotional trajectory information through sequential processing. It does not treat the history as an undifferentiated bag of turns. The gate mechanism of the LSTM naturally allows it to selectively retain or forget information from earlier turns based on relevance to the current prediction, which is functionally similar to (though less explicit than) modeling emotional transitions directly.
> >
> > We note that our new Transformer comparison experiment (see our response to Reviewer HXnf) provides indirect evidence on this point: the Transformer's explicit long-range attention patterns do not improve over the LSTM's sequential processing, suggesting that local emotional dynamics captured by the recurrent state are sufficient for the task.

---

### Author Response · Authors · 2026-02-26
**Summary of New Evidence and Planned Revisions**

We sincerely thank all three reviewers for their thorough and constructive feedback. The reviews have helped us identify important areas for improvement, and we have conducted substantial new experiments to address the concerns raised. Below we provide a point-by-point response to each reviewer, along with a summary of new evidence and planned revisions.

All new experiments follow the same rigorous protocol as the original submission: 10 random seeds, frozen Sentence-RoBERTa encoder, paired statistical testing, and identical hyperparameters.

### New Experiments Completed

All experiments use 10 random seeds with the same protocol as the original submission.

| Experiment | Scale | Key Finding |
|-----------|-------|-------------|
| MELD cross-validation | 60 runs (6 conditions × 10 seeds) | Core IEMOCAP patterns replicate on a structurally different dataset |
| Transformer vs. LSTM | 80 runs (8 conditions × 10 seeds) | Transformer ≤ LSTM; saturation is not an LSTM artifact |
| Discourse marker confounds | 5 mixed-effects models | Sad effect survives all controls; Angry/Happy were length confounds |

### Revisions Committed

| Category | Changes |
|----------|---------|
| Title | Remove 'for Generation'; adopt more accurate title |
| Figure 3 | Reframe around saturation (K=10 to 30); replace peak K with saturation K in Figure 3(b) |
| Section 3.7 | Add explicit Figure 3 reference; acknowledge K=0 baseline strength |
| Section 4.2 | Expand discussion of Cramér's V implications; narrow claims to Sad |
| Section 3.8 | Scope SenticNet conclusion to simple integration methods |
| Section 4.4 | Add text-based features for Happy/Excited distinction |
| Section 5 | Add Broader Impact paragraph; expand generalizability discussion |
| References | Fix Table ?? broken references (two instances); minor formatting |
| New content | MELD cross-validation subsection, Transformer comparison ablation, confound analysis results |

---

### Decision · Action_Editor_QBnC · 2026-05-14

**Recommendation:** Accept with minor revision

**Additional Comments:**

The paper requires some minor edits in the references to make them more consistent. Some venue names are only mentioned as abbreviations while some contain the full name; some have the URL (remove), etc.

**Audience:**

Yes

**Audience Explanation:**

The paper is relevant to researchers in affective computing and conversational NLP.

**Claims And Evidence:**

Yes

**Claims Explanation:**

The paper presents a study with convincing evidence, statistical evaluation, and useful insights into context saturation, emotion-specific dependencies, and discourse-level patterns in emotion recognition in conversations.

---

> ### Author Response · Authors · 2026-06-10
> **Final Submission!**
>
> We made the minor camera-ready edits requested by the action editor. Specifically, we standardized venue names across the bibliography, removed unnecessary URL fields from reference entries, checked capitalization and metadata consistency, and updated the citation information to match the accepted version.
>
> We also added a code availability statement and prepared a public reproducibility repository at https://github.com/phillelenina/causal-erc-context-saturation. We updated the paper metadata and author information for the camera-ready version.
>
> Thank you so much!